# Watch and Listen: Understanding Audio-Visual-Speech Moments with Multimodal LLM

**Zinuo Li[1], Xian Zhang[1], Yongxin Guo[2], Mohammed Bennamoun[1], Farid Boussaid[1], Girish Dwivedi[1], Luqi Gong[3]\*, Qiuhong Ke[4]\***

https://github.com/zinuoli/TriSense

[1]University of Western Australia [2]Alibaba Group [3]Zhejiang Laboratory [4]Monash University

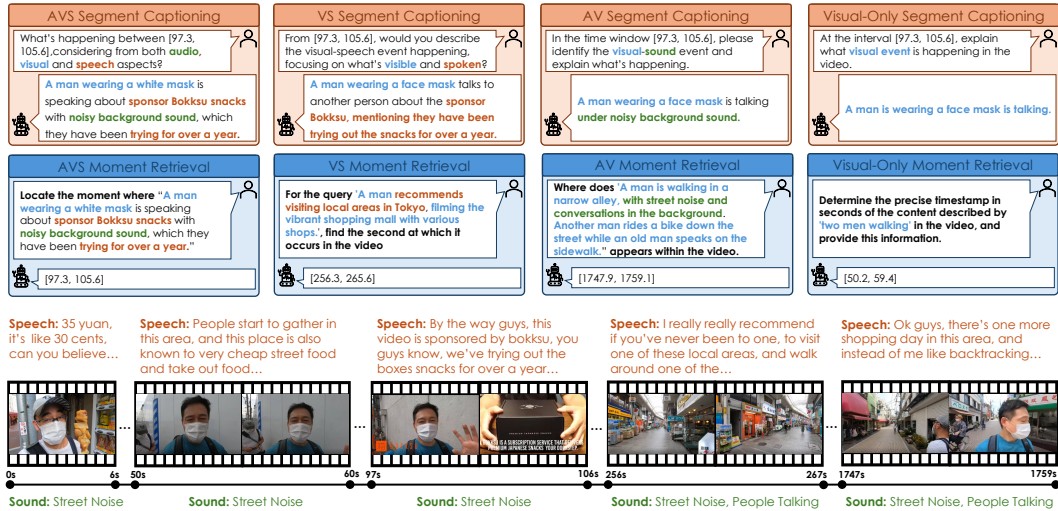

Figure 1: TriSense supports segment captioning and moment retrieval for videos from audio, visual, and speech modalities, as well as any combination of them, covering a total of eight different tasks.

## Abstract

Humans naturally understand moments in a video by integrating visual and auditory cues. For example, localizing a scene in the video like *"A scientist passionately speaks on wildlife conservation as dramatic orchestral music plays, with the audience nodding and applauding"* requires simultaneous processing of visual, audio, and speech signals. However, existing models often struggle to effectively fuse and interpret audio information, limiting their capacity for comprehensive video temporal understanding. To address this, we present **TriSense**, a triple-modality large language model designed for holistic video temporal understanding through the integration of visual, audio, and speech modalities. Central to TriSense is a **Query-Based Connector** that adaptively reweights modality contributions based on the input query, enabling robust performance under modality dropout and allowing flexible combinations of available inputs. To support TriSense's multimodal capabilities, we introduce **TriSense-2M**, a high-quality dataset of over 2 million curated samples generated via an automated pipeline powered by fine-tuned LLMs. TriSense-2M includes long-form videos and diverse modality combinations, facilitating broad generalization. Extensive experiments across multiple benchmarks demonstrate the effectiveness of TriSense and its potential to advance multimodal video analysis.

---

\*corresponding authors

39th Conference on Neural Information Processing Systems (NeurIPS 2025).

# 1 Introduction

Human understanding of real-world events is inherently multimodal: we rely not only on vision but also on spoken language and audio cues to make sense of what is happening in a video. This integration allows us to interpret intention, emotion, and the significance of events—going beyond what is seen to include what is heard and said. However, existing Multimodal Large Language Models (MLLMs) often fall short of achieving this level of nuanced video temporal understanding. While advances in vision-language modeling and temporal localization have improved language–visual alignment [11, 23, 15, 12, 38, 19, 28], most models still rely solely on visual inputs. As a result, they perform poorly on tasks requiring the integration of audio and speech—particularly when one or more modalities are missing, noisy, or contextually irrelevant. This stands in contrast to human perceptual robustness and significantly limits model generalization in real-world scenarios, such as accurately localizing events or generating rich multimodal descriptions.

**Challenges and Current Limitations.** As highlighted above, the current state of multimodal temporal understanding remains limited. Building on these observations, two core challenges continue to hinder progress in multimodal temporal understanding: **1) Insufficient and Incomplete Training Data:** Current datasets are often composed of short clips and lack large-scale, fully annotated examples across all three modalities—vision, audio, and speech—which are essential for effective multimodal pretraining [14, 8, 13, 12]. This scarcity hampers the development of robust MLLMs. Moreover, real-world videos often contain incomplete or inconsistent modality coverage, due to factors like variable recording setups, intentional omissions (e.g., silent footage or background music), or the natural absence of certain signals in specific scenes. When models are trained predominantly on videos with all modalities present, they often fail at inference time when confronted with missing or degraded inputs—a common scenario in the wild. **2) Lack of Modality Adaptation:** Current MLLMs are generally not equipped to assess the relative importance of each modality based on task or query context. Recent models such as LongVALE [10] and Qwen2.5-Omni [34] attempt to integrate multiple modalities but fall short in adaptivity. For instance, LongVALE compresses all modality tokens into a single representation, resulting in information loss and poor handling of missing modalities. It also lacks an adaptive dropout strategy, leading to unstable performance when modality availability varies. Qwen2.5-Omni introduces temporal positional embeddings, but still fails to capture fine-grained temporal dependencies, limiting its effectiveness on complex moment-level tasks, as demonstrated in our experiments.

**Key Contributions.** We argue that understanding complex moments in video requires not only broader modality coverage but also an adaptive mechanism to selectively emphasize the most relevant modalities depending on the task and query. Our approach addresses these challenges through the following key contributions:

**1)** We introduce *TriSense-2M*, a large-scale multimodal dataset containing 2 million annotations. Each video instance in the dataset includes event-based annotations across vision, audio, and speech modalities, with **flexible combinations** and natural absences of modalities. The dataset supports a wide variety of scenes and includes long-form videos averaging 905 seconds—significantly longer than those in existing datasets—enabling deeper and more realistic temporal understanding. Importantly, queries are expressed in high-quality natural language, aligned with temporal annotations, and span diverse modality configurations to facilitate robust multimodal learning.

**2)** We propose *TriSense*, a triple-modality MLLM designed for both video segment captioning and moment retrieval under diverse modality configurations. As depicted in Figure 1, TriSense is designed to handle multimodal video data with varying availability of vision, audio, and speech over temporal dimension. Crucially, it includes a **Query-Based Connector** that dynamically adjusts modality weights based on the query's content and context. This allows the model to emphasize the most informative modalities (e.g., prioritizing vision if most relevant) while down-weighting irrelevant or missing ones—enabling strong performance even under incomplete modality conditions.

**3)** We conduct extensive experiments on two core tasks—video segment captioning and moment retrieval—across eight modality configurations, including zero-shot evaluation on public benchmarks. TriSense achieves strong performance on **both** the new TriSense-2M dataset and existing benchmarks, laying a solid foundation for future research in multimodal temporal video understanding.

## 2 Related Work

### 2.1 Video Temporal Understanding MLLMs

Video temporal understanding focuses on modeling how events unfold over time within a video, enabling tasks such as moment retrieval, segment captioning, and dense video captioning. Vision-language models (VLMs) have demonstrated strong capabilities in solving real-world problems, including in zero-shot scenarios without task-specific fine-tuning. However, many of these models still face challenges when it comes to understanding temporal dynamics [36, 5]. To address this, several models have been fine-tuned on video grounding datasets that emphasize temporal structure—such as TimeChat [27], VTimeLLM [15]—to enhance their temporal reasoning abilities. More recently, Momentor [23] introduced a time encoder to correct errors caused by time token quantization, while VTG-LLM [12] employed specialized time tokens and temporal position embeddings to help video LLMs better capture the timing of events. In a different approach, TRACE [11] applied principles from causal language modeling to propose causal event modeling for videos. It also introduced a lightweight time tower for encoding temporal information, achieving solid performance in temporal understanding.

Extending beyond vision-language modeling, Multimodal Large Language Models (MLLMs) integrate visual, audio, and speech modalities to enable richer video analysis. To improve performance, recent efforts have focused on incorporating these additional modalities. For example, LongVALE [10] compresses all modality tokens into a single token, but this leads to loss of fine-grained details such as audio context or speech intonation. Qwen2.5-Omni [34], on the other hand, introduces the Thinker-Talker architecture—a fully end-to-end multimodal system that handles text, images, audio, and video inputs. To better align audio and video along the temporal dimension, the authors propose TMRoPE, a timing-aware positional encoding technique. However, our experiments show that this model still struggles with grounding in long-form videos.

Despite these advancements, many existing models are either limited to visual modalities or lack support for flexible combinations of different modalities. This restricts their effectiveness in temporal understanding tasks, particularly in scenarios where some modalities may be absent or noisy. These challenges motivate our pursuit of MLLMs that can robustly handle various modality combinations while maintaining strong temporal reasoning and general understanding performance.

### 2.2 Benchmarks for Temporal Understanding

The development of benchmark datasets has played a crucial role in advancing video temporal understanding. Early contributions such as DiDeMo [14] introduced natural language queries for moment localization in videos. Subsequent datasets like Charades-STA [8] and ActivityNet Captions [18] expanded on this by covering a wider variety of actions and longer video durations, significantly pushing the field forward. More recently, InternVid2 [32] has emerged as a large-scale foundational dataset, offering 61 million audio-visual-speech annotations. However, many of these annotations are disjointed, lacking coherence across modalities, and the dataset contains a substantial number of low-quality captions due to its scale.

To address these limitations, VAST-27M [3] and VALOR [2] were introduced. Both datasets offer high-quality, omni-modal (audio-visual-speech) annotations with better interrelated features than InternVid2, supporting comprehensive multimodal understanding for MLLMs. Nonetheless, they rely on simple concatenation of captions across modalities and do not incorporate cross-modal reasoning. Moreover, these benchmarks provide only coarse-grained captions for short clips, making them inadequate for fine-grained understanding of long-form video. Although both datasets improve on InternVid2, they repeat similar limitations: their annotations are not contextually integrated across modalities, and their temporal granularity is too coarse for modeling nuanced event transitions in extended videos. In response to these shortcomings, LongVALE [10] was proposed, featuring 108K high-quality annotations with omni-modal coverage. While it offers notable improvements, VAST-27M, VALOR, and LongVALE all overlook a critical issue: the dynamic presence of modalities. In real-world videos, audio, visual, and speech inputs are not always available simultaneously, raising important questions about model robustness in the face of missing modalities.

In conclusion, existing benchmarks often focus exclusively on visual information or fail to adequately support flexible multimodal integration. These limitations highlight the need for improved datasets and serve as a key motivation for our work.

## 3 Data Construction

As discussed in Sections 1 and 2, while some existing datasets include all three modalities—visual, audio, and speech—they generally assume that these modalities are always present simultaneously [3, 2, 10], overlooking the importance of supporting arbitrary combinations. This assumption limits the development of models that can handle missing or partial inputs effectively. To address this, we introduce a new large-scale, high-quality dataset designed to support both fully multimodal and partially multimodal scenarios.

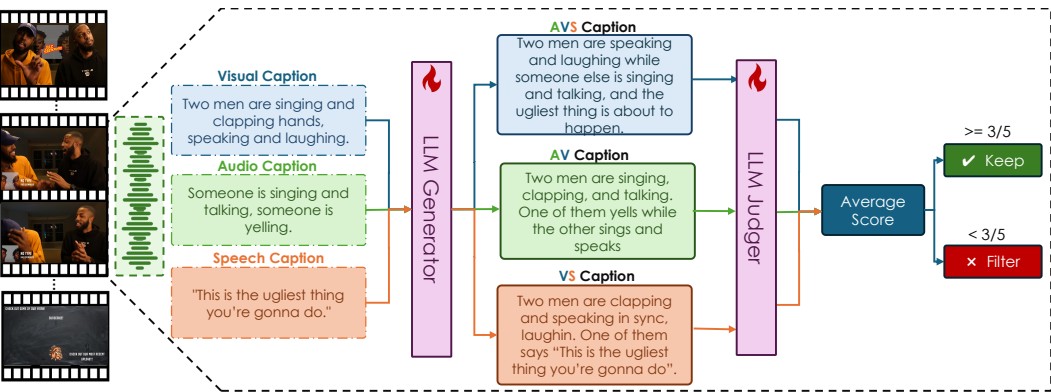

Figure 2: We employ an automated framework to build our dataset by leveraging modality-specific captions from vision, audio, and speech streams. Two large language models (LLMs) are trained for this process: a **Generator**, which fuses the three input captions into multi-modal outputs (AVS, AV, VS), and a Judger, which evaluates the semantic quality of the generated captions. The **Judger** assigns an average quality score between 0 and 5 based on alignment with the original inputs. Samples scoring ≥ 3 are retained, while those scoring < 3 are discarded.

Our dataset includes longer video durations, making it suitable for realistic and fine-grained temporal grounding tasks. It enables models to "watch and listen" in a human-like manner, flexibly attending to available visual, auditory, and speech cues to identify relevant moments. In addition, we include caption data to promote deeper video understanding and narrative generation. To support scalable and consistent annotation, we developed a fully automated data construction pipeline, as shown in Figure 2.

We begin by selecting subsets from InternVid [32] and VAST [3] as our raw sources for both video content and initial captions. Each video clip is annotated with three distinct captions: a visual caption that describes observable scenes and actions, an audio caption that details acoustic elements, and a speech caption that transcribes spoken content. These modality-specific captions are generated using specialized annotation pipelines adapted from previous works [32, 3].

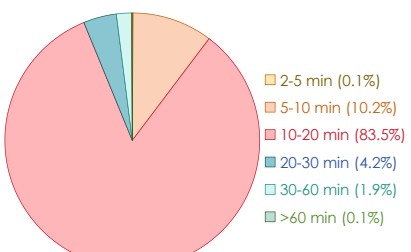

Figure 3: Video duration distribution. Most videos are 10–20 minutes long (83.5%), supporting long-form temporal understanding.

To enable reasoning across modalities, our goal is to synthesize omni-modal captions that flexibly combine distinct unimodal annotations. These are crucial for training models capable of comprehensive temporal understanding. To generate these captions, we use two custom-trained large language models based on Qwen2.5-72B [35]: a **Generator** and a **Judger**. The Generator merges modality-specific captions into unified representations for three combinations: AVS (Audio-Visual-Speech), AV (Audio-Visual), and VS (Visual-Speech). These captions are designed to capture cross-modal interactions, such as clapping that aligns with vocal rhythm or speech that corresponds with visual context. The Judger evaluates the quality of each synthesized caption by measuring its semantic alignment with the original unimodal annotations. It assigns a quality score ranging from 0 to 5 and filters out samples with inconsistencies, such as speech that does not relate to visual actions or mismatched audio-visual descriptions. To train these models, we first build a high-quality reference corpus using GPT-o1 [16], which is then

**manually** refined and filtered. From this curated set, we select 10,000 samples to train the Generator and 3,000 samples to train the Judger. Further training details are provided in the Appendix.

Table 1: **Comparison of temporal understanding datasets.** TriSense-2M uniquely supports all three modalities with long video lengths and explicit handling of modality dropout.

| Datasets | Annotations | Avg. len | Visual | Audio | Speech | Modality Dropout |
|---|---|---|---|---|---|---|
| VALOR [2] | 1.32M | 10s | ✓ | ✓ | ✗ | ✗ |
| VAST [3] | 27M | 30s | ✓ | ✓ | ✗ | ✗ |
| UnAV-100 [9] | 30K | 42.1s | ✓ | ✓ | ✗ | ✗ |
| Charades-STA [8] | 16K | 30s | ✓ | ✗ | ✗ | ✗ |
| ActivityNet-Captions [13] | 20K | 180s | ✓ | ✗ | ✗ | ✗ |
| LongVALE [10] | 108K | 235s | ✓ | ✓ | ✓ | ✗ |
| TriSense-2M | 2M | 905s | ✓ | ✓ | ✓ | ✓ |

Table 2: **Detailed Information of TriSense-2M**, where Avg/Min/Max Duration represent the average, minimum, and maximum duration, respectively. 0–5s / 5–10s / 10–15s, etc., represent the proportions of different duration intervals.

| Total Events | Avg Duration | Min Duration | Max Duration | 0-5s | 5-10s | 10-15s | 15-20s | 20-30s |
|---|---|---|---|---|---|---|---|---|
| 1940522 | 6.87s | 2.00s | 30.00s | 35.5% | 46.2% | 12.6% | 4.2% | 1.5% |

The data construction pipeline processes an initial pool of 5 million multimodal video samples containing visual, audio, and speech captions. Through multiple rounds of generation, evaluation, and filtering, the Judger retains only high-quality outputs, resulting in a final dataset of 2 million samples drawn from approximately 38,000 long videos. The distribution of video durations is shown in Figure 3. The average video length is 905 seconds, nearly four times longer than that of the closest existing dataset, which averages 235 seconds [10]. This curated dataset enables robust temporal reasoning across diverse modality combinations and forms the foundation for training and evaluating our TriSense model. Overall comparisons with existing datasets are provided in Table 1, and more detailed examples are included in the **Appendix.**

## 4 TriSense Architecture

The overall architecture of the TriSense model is illustrated in Figure 4. The model is designed to process visual, audio, and speech information extracted from a video in order to answer a text-based query. Each modality is first processed by one of three specialized expert encoders [24, 26, 4]. The resulting feature representations are then passed through modality-specific projectors and integrated with the query using Cross-Attention mechanisms, allowing the model to capture fine-grained interactions between each modality and the query. A Query-Based Connector subsequently fuses the modality representations and adaptively adjusts their contributions based on the content of the query. This fused multimodal representation is enriched with temporal information and then input into a Large Language Model (LLM) backbone, which produces the final output. To effectively model temporal dependencies, the architecture includes a dedicated Time Encoder [11] and employs task-specific heads to ensure outputs are temporally aligned and task-relevant.

### 4.1 Multimodal Information Extraction

During training, given a video $V = \{F_1, F_2, ....F_T\}$, where $T$ denotes the total number of frames, we first uniformly select $n = 64$ frames for further processing and controlling memory usage. For each selected frame, we record its timestamp and extract an audio segment of $\pm 1$ second around the frame, resulting in a sequence of audio segments $A = \{A_1, A_2, ....A_n\}$, each lasting 2 seconds. For instance, if a frame is sampled at 123.4 seconds, the corresponding audio segment spans from 122.4 to 124.4 seconds. We then use pretrained expert encoders to extract modality-specific tokens: $f_i^v$, $f_i^a$, $f_i^s$ for vision, audio, and speech, repectively.

To enhance temporal awareness of the model, we encode the selected timestamp using a Time Encoder [11]. Each timestamp is first tokenized into a fixed-length character sequence comprising four integer digits, a decimal point, and one fractional digit, resulting in 6 tokens per frame. For example, the timestamp [123.4] is tokenized to $\langle 0 \rangle \langle 1 \rangle \langle 2 \rangle \langle 3 \rangle \langle . \rangle \langle 4 \rangle$. These tokens are then embedded to form time features $f(t)$. Given the high token count from the three modalities, we apply Slot-Based

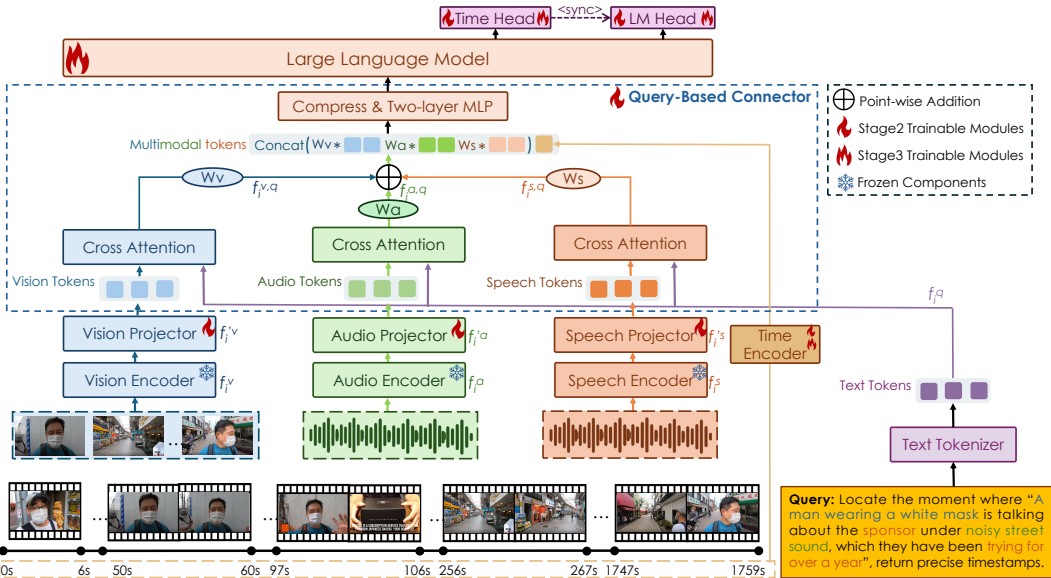

Figure 4: **Architecture of the TriSense model.** The model processes vision, audio, and speech via dedicated encoders and fuses them using a Query-Based Connector that assigns weights based on the query. The fused output, combined with temporal embeddings, is passed to an LLM for generating timestamped or textual responses.

Compression [12] as Modality Projector to reduce the dimensionality of the input. This technique compresses the vision tokens $f_i^v$, audio tokens $f_i^a$, and speech tokens $f_i^s$, into fixed-length vectors of 16 tokens each, denoted as $f_i^{'v}$, $f_i^{'a}$ and $f_i^{'s}$.

## 4.2 Query-Based Connector

To more effectively integrate multimodal inputs with the query and enhance sensitivity to salient features, we introduce a Query-Based Connector that adaptively balances the contributions of each modality based on the query's content, as illustrated in Figure 4. The compressed modality features $f_i^{'v}$, $f_i^{'a}$ and $f_i^{'s}$ obtained from Section 4.1, are passed through Cross-Attention layers, where they interact with the encoded query representation $f^{(q)}$. The objective is for each attention layer to emphasize features that are most relevant to the query. The outputs of these layers are denoted as query-relevant features $f_i^{v,q}$, $f_i^{a,q}$ and $f_i^{s,q}$, which reflect the alignment between each modality and the query.

To dynamically determine the importance of each modality in relation to the query, we introduce an adaptive weighting mechanism. First, we apply global average pooling over the sequence dimension of each modality to derive compact global representations $c_v$, $c_a$, and $c_s$. These vectors are concatenated and fed into a single-layer MLP $\mathcal{F}(\cdot)$ to generate unnormalized weights $w_v$, $w_a$, and $w_s$. The weights are then normalized using a softmax function to yield a valid probability distribution over the modalities, satisfying the constraint $w_v + w_a + w_s = 1$. The computation is formalized below:

$$\mathbf{c}_m = \frac{1}{T_m} \sum_{t=1}^{T_m} \mathbf{x}_{m,t}, \qquad m \in \{v, a, s\} \tag{1}$$

$$\tilde{w}_m = \mathcal{F}\big([\, c_v \,\|\, c_a \,\|\, c_s \,]\big) \in \mathbb{R}^3, \qquad m \in \{v, a, s\} \tag{2}$$

$$w_m = \frac{\exp(\tilde{w}_m)}{\displaystyle\sum_{m' \in \{v,a,s\}} \exp(\tilde{w}_{m'})}, \qquad m \in \{v, a, s\} \tag{3}$$

Here, $\mathbf{c}_m$ is the compressed modality vector after global average pooling, $T_m$ indicates the sequence length of distinct modality $m$, $\|$ denotes vector concatenation, and $\tilde{w}_m$ and $w_m$ represent unnormalized and normalized weights, respectively.

After computing the weights, we multiply the previously obtained query-relevant features by their corresponding weights and concatenate them together to fuse a multimodal representation. To reduce the token count, we apply slot compression $C_{comp}(\cdot)$ again to compress the tokens from triple the amount back to a single scale. Finally, a two-layer MLP $\hat{\mathcal{F}}(\cdot)$ is used for further feature

refinement, aligning the representation with the LLM's input dimensionality and enhancing its expressive capacity:

$$\mathcal{X}_m = \hat{\mathcal{F}}(C_{comp}(\text{concat}(w_v f_i^{v,q}, w_a f_i^{a,q}, w_s f_i^{s,q}))) \qquad (4)$$

This adaptive fusion allows the model to emphasize the most informative modalities while reducing the influence of less relevant ones, based on the specific query. The resulting representation $\mathcal{X}_m$ is combined with the corresponding time embeddings $f(t)$ (as introduced in Section 4.1) and passed into the LLM backbone, which uses its contextual reasoning capabilities to generate the final output.

### 4.3 Causal Event Prediction

To enhance the model's temporal reasoning capabilities and better align predictions with the underlying structure of video narratives, we employ causal event prediction, a method shown to be effective in prior work [11]. This approach enables the model to reason about cause-and-effect relationships across time, predicting upcoming events based on prior context. Specifically, given a video V, we segment it into a sequence of events $\{e_1, e_2, \cdots, e_K\}$, where each event $e_k = (t_k, c_k)$ consists of a timestamp $t_k$ and an associated caption $c_k$ describing the video segment:

$$\mathbf{V} = \{e_1, e_2, \cdots, e_k\} = \{(t_k, c_k) | 1 \leq k \leq K\}. \qquad (5)$$

Our goal is to predict the next event $e_k$ conditioned on the sequence of prior events $e_{1:k-1}$, the user-provided query $\mathbf{Q}$, and the multimodal features $\mathcal{X}_m$ produced by the Query-Based Connector:

$$\mathcal{P}(e_k | e_{1:k-1}, f^{(q)}, \mathcal{X}_m) = \mathcal{P}(t_k, c_k | e_{1:k-1}, f^{(q)}, \mathcal{X}_m) \qquad (6)$$

To support both temporal and textual outputs, we introduce adaptive head switching via a special $\langle\text{sync}\rangle$ token during LLM generation. This token is appended to the vocabulary and serves as a control signal that guides the model to switch between time head and language model (LM) head, as illustrated in Figure 4. When the $\langle\text{sync}\rangle$ token is encountered, the LLM transitions between decoding modalities to generate either timestamp-aligned predictions or free-form textual outputs, depending on the task.

## 5 Experiments

In this section, we present the core experiments conducted to evaluate the performance of our proposed model. Due to space constraints, implementation details, training procedures, and additional experimental results are provided in the **Appendix.**

### 5.1 Evaluation Datasets, Metrics and Baseline Models

To rigorously assess the effectiveness of TriSense, we conduct evaluations on two key temporal understanding tasks:

- **Segment Captioning (SC)**. This task involves generating descriptive captions that accurately summarize the events occuring throughout a video. We evaluate our model on the newly introduced TriSense-2M dataset, and provide additional datasets in the Appendix. The performances are reported in BLEU-4 [22], CIDEr [30], ROUGE_L [20] and METEOR [1] to gauge the quality and accuracy of the generated captions.
- **Moment Retrieval (MR)**. In this task, the model is required to retrieve specific segments within a video that correspond to a given textual query. We evaluate performance on TriSense-2M, as well as two widely used public benchmarks: Charades-STA [8] and ActivityNet-Captions [13]. Retrieval effectiveness is reported using Recall@IoU=0.5, Recall@IoU=0.7, and mean IoU (mIoU), providing a comprehensive view of the model's localization accuracy at varying overlap thresholds.

The modality combinations are set to Audio-Visual-Speech (AVS), Visual-Speech (VS), Audio-Visual (AV) and Visual-Only (V). To establish a solid comparative baseline, we select representative models specifically designed for Video Temporal Grounding (VTG) tasks. These include VTimeLLM [15], TimeChat [27], VTG-LLM [12], TRACE [11], along with two recent omni-modal models Long-VALE [10] and Qwen2.5-Omni [34]. While Momentor [23], Hawkeye [31] and NumPro-FT [33] are not included in the TriSense-2M benchmark due to the unavailability of model checkpoints and their lack of support for captioning, they are included in public benchmark evaluations according to the

Table 3: **Segment Captioning results.** Performance is reported on four modality settings using BLEU-4 (B), METEOR (M), ROUGE-L (R), and CIDEr (C). Best and second-best results are in **bold** and underlined, respectively.

| Model | AVS-SC | | | | VS-SC | | | | AV-SC | | | | V-SC | | | |
|---|---|---|---|---|---|---|---|---|---|---|---|---|---|---|---|---|
| | B | M | R | C | B | M | R | C | B | M | R | C | B | M | R | C |
| VTimeLLM (7B) | 0.8 | 8.2 | 16.1 | 2.4 | 1.2 | 8.8 | 16.9 | 3.1 | 1.3 | 10.3 | 17.9 | 2.6 | 1.4 | 10.4 | 18.2 | 4.0 |
| TimeChat (7B) | 0.6 | 4.0 | 8.7 | 0.6 | 0.9 | 4.9 | 9.8 | 1.4 | 1.1 | 5.5 | 10.5 | 1.5 | 0.8 | 6.7 | 12.5 | 5.7 |
| VTG-LLM (7B) | 0.3 | 4.8 | 9.6 | 0.6 | 0.3 | 4.9 | 10.0 | 0.9 | 0.4 | 5.2 | 10.2 | 0.9 | 0.3 | 5.0 | 9.8 | 1.4 |
| TRACE (7B) | 1.0 | 7.6 | 13.5 | 1.1 | 1.4 | 7.8 | 14.3 | 2.3 | 1.6 | 9.0 | 16.3 | 2.6 | 1.3 | 9.4 | 16.8 | 9.5 |
| TRACE-uni (7B) | 1.1 | 8.2 | 14.7 | 1.4 | 1.5 | 8.3 | 15.1 | 2.2 | 1.6 | 9.5 | 16.3 | 2.3 | 1.3 | 9.9 | 17.6 | 8.8 |
| LongVALE (7B) | 1.2 | 8.6 | 16.7 | 4.9 | 2.3 | 10.0 | 20.1 | 5.5 | 2.5 | 11.4 | 21.3 | 5.9 | 1.5 | 11.5 | 18.8 | 0.9 |
| Qwen2.5-Omni (7B) | 0.8 | 8.8 | 13.1 | 1.7 | 0.8 | 8.6 | 13.1 | 0.8 | 1.2 | 9.8 | 15.1 | 1.3 | 1.1 | 10.1 | 14.6 | 1.1 |
| TriSense (7B) | **3.4** | **10.1** | **20.1** | **8.3** | **3.0** | **10.0** | **22.2** | **11.8** | **5.3** | **12.2** | **26.3** | **15.4** | **7.3** | **12.6** | **30.7** | **36.3** |

Table 4: **Moment Retrieval results with 64 frames.** Performance is reported as Recall at IoU 0.5 and 0.7 across four modality settings. Best and second-best results are in **bold** and underlined, respectively.

| Model | AVS-MR | | VS-MR | | AV-MR | | V-MR | |
|---|---|---|---|---|---|---|---|---|
| | IoU=0.5 | IoU=0.7 | IoU=0.5 | IoU=0.7 | IoU=0.5 | IoU=0.7 | IoU=0.5 | IoU=0.7 |
| VTimeLLM (7B) | 0.21 | 0.09 | 0.28 | 0.14 | 0.23 | 0.08 | 0.41 | 0.14 |
| TimeChat (7B) | 0.28 | 0.12 | 0.27 | 0.09 | 0.22 | 0.08 | 0.34 | 0.12 |
| VTG-LLM (7B) | 0.19 | 0.08 | 0.15 | 0.05 | 0.21 | 0.07 | 0.23 | 0.06 |
| TRACE (7B) | 0.39 | 0.12 | 0.31 | 0.15 | 0.24 | 0.13 | 0.42 | 0.21 |
| TRACE-uni (7B) | 0.30 | 0.17 | 0.35 | 0.17 | 0.24 | 0.18 | **0.48** | **0.22** |
| LongVALE (7B) | 0.08 | 0.01 | 0.07 | 0.01 | 0.07 | 0.01 | 0.05 | 0.01 |
| Qwen2.5-Omni (7B) | 0.61 | 0.21 | 0.61 | 0.16 | 0.28 | 0.07 | 0.18 | 0.06 |
| TriSense (7B) | **1.12** | **0.42** | **0.80** | **0.28** | **0.57** | **0.21** | 0.43 | **0.22** |

reports in their official papers. By comparing our model against these baselines across diverse tasks and evaluation metrics, we aim to provide a comprehensive assessment of TriSense's capabilities and its advancements in video temporal understanding.

## 5.2 Results and Analysis

**Superior Performance on Omni-Modal datasets.** We evaluate our performance on the proposed TriSense-2M dataset. As illustrated in Table 3 and Table 4, TriSense consistently outperforms existing video LLMs across nearly all evaluated tasks. It also significantly surpasses latest omni-modal models like LongVALE [10] and Qwen2.5-Omni [34], particularly in the audio-visual-speech (AVS) setting where all three modalities are leveraged.

We observe that the model shows slightly lower performance on visual-only moment retrieval compared to state-of-the-art vision models, which is likely due to its optimization for multimodal settings rather than visual-only scenario. Also, it is important to note that our model uses only 64 input frames during testing, compared to larger input sizes used in other models—such as 128 frames in TRACE [11] and 100 frames in VTimeLLM [15]. Since TriSense-2M consists mostly of long videos, using fewer frames makes it more difficult for the model to achieve high accuracy in long-video moment retrieval tasks. When the videos become shorter or more frames are used, we gain better performances, this is also supported by the results in Table 6 and Table 7.

We also conduct experiments on the public Omni-Modal benchmark LongVALE [10]. LongVALE is designed for event understanding across vision, audio, and language modalities, comprising 105,000 omni-modal events with precise temporal annotations and relation-aware captions, collected from 8,400 high-quality long-form videos. The Omni-VTG and Omni-SC tasks named from the official report of LongVALE, which are the same as AVS-MR and AVS-SC in our paper. As summarized in Table 5, our zero-shot performance on the Moment Retrieval task is comparable to LongVALE's performance, even though their model is trained on the same dataset. Although there is a larger gap in the Segment Captioning task, we believe this is due to pattern differences in captioning styles between our Segment Captioning training data and LongVALE's Segment Captioning data. Such differences in caption patterns can lead to noticeable drops across all four evaluation metrics.

Table 5: Performance on public Omni-Modal benchmark LongVALE [10]. **"*"** indicates this model is trained on the LongVALE dataset. The best and second results are highlighted in bold and underlined, respectively.

| Model | Omni-VTG (AVS-MR) | | | | Omni-SC (AVS-SC) | | | |
|---|---|---|---|---|---|---|---|---|
| | R@0.3 | R@0.5 | R@0.7 | mIoU | B | M | R | C |
| VideoChat (7B) | 2.2 | 0.9 | 0.4 | 3.0 | 0.5 | 9.6 | 0.0 | 8.2 |
| VideoChatGPT (7B) | 4.9 | 2.9 | 0.9 | 5.0 | 0.4 | 14.0 | 0.9 | 5.9 |
| VideoLLaMA (7B) | 2.5 | 1.1 | 0.3 | 1.9 | 0.9 | 11.5 | 0.1 | 8.9 |
| PandaGPT (7B) | 2.5 | 1.0 | 0.3 | 2.2 | 0.6 | 14.9 | 0.3 | 8.9 |
| NExT-GPT (7B) | 4.3 | 1.9 | 0.7 | 4.0 | 0.4 | 10.2 | 0.0 | 8.1 |
| TimeChat (7B) | 5.8 | 2.6 | 1.1 | 5.2 | 1.2 | 16.1 | 1.6 | 10.0 |
| VTimeLLM (7B) | 7.5 | 3.4 | 1.3 | 6.4 | 1.0 | 14.5 | 1.6 | 5.5 |
| LongVALE* (7B) | **15.7** | 8.6 | 3.9 | 11.0 | **5.6** | **22.4** | **20.3** | **10.9** |
| TriSense (7B) | 14.8 | **9.3** | **4.7** | **11.2** | 4.8 | 21.9 | 18.8 | 10.4 |

Table 6: **Zero-shot Moment Retrieval results on public benchmarks with 64 frames.** "*" indicates this model uses more frames than TriSense. The top and second-best results are highlighted in **bold** and underlined, respectively.

| Model | Charades-STA | | | ActivityNet-Caption | | |
|---|---|---|---|---|---|---|
| | IoU=0.5 | IoU=0.7 | mIoU | IoU=0.5 | IoU=0.7 | mIoU |
| VTimeLLM* (7B) | 27.5 | 11.4 | 31.2 | 27.8 | 14.3 | 30.4 |
| VTimeLLM* (13B) | 34.3 | 14.7 | 34.6 | 29.5 | 14.2 | 31.4 |
| TimeChat* (7B) | 32.2 | 13.4 | - | 4.6 | 2.0 | 6.9 |
| Momentor* (7B) | 26.6 | 11.6 | 28.5 | 23.0 | 12.4 | 29.3 |
| HawkEye (7B) | 31.4 | 14.5 | 33.7 | 29.3 | 10.7 | 32.7 |
| VTG-LLM* (7B) | 33.8 | 15.7 | - | 8.3 | 3.7 | 12.0 |
| TRACE* (7B) | 40.3 | 19.4 | 38.7 | 37.7 | 24.0 | 39.0 |
| TRACE-uni* (7B) | **43.7** | 21.0 | **41.5** | 38.2 | 24.7 | 39.4 |
| NumPro-FT* (7B) | 42.0 | 20.6 | 41.4 | 37.5 | 20.6 | 38.8 |
| TriSense (7B) | 42.3 | **27.6** | 39.8 | **39.6** | **27.2** | **40.1** |

**Zero-shot performance on public Moment Retrieval benchmarks.** We also evaluate TriSense in a zero-shot setting on established two classical visual-only datasets, including Charades-STA [8] and ActivityNet-Captions [13], as shown in Table 6. The results show that although TriSense shows slight inferior performance in Table 4, it still achieves competitive performance in visual-only settings, showing especially **higher accuracy (IoU=0.7)** than others, even with **less frames** used.

Table 7: **Ablation studies on Moment Retrieval.** The top and second-best results are highlighted in **bold** and underlined, respectively.

| Model | Frame Number | AVS-MR | | VS-MR | | AV-MR | | V-MR | |
|---|---|---|---|---|---|---|---|---|---|
| | | IoU=0.5 | IoU=0.7 | IoU=0.5 | IoU=0.7 | IoU=0.5 | IoU=0.7 | IoU=0.5 | IoU=0.7 |
| *Training Stages* | | | | | | | | | |
| Stage1 Only | 64 | 0.07 | 0.01 | 0.06 | 0.01 | 0.06 | 0.00 | 0.02 | 0.00 |
| Stage1+2 | 64 | 0.52 | 0.19 | 0.43 | 0.18 | 0.32 | 0.12 | 0.27 | 0.14 |
| *Connector* | | | | | | | | | |
| Addition | 64 | 0.71 | 0.22 | 0.69 | 0.21 | 0.41 | 0.11 | 0.22 | 0.19 |
| Fixed Weights | 64 | 0.89 | 0.38 | 0.77 | 0.24 | 0.52 | 0.19 | 0.44 | 0.23 |
| *Frame Number* | | | | | | | | | |
| TriSense (7B) | 32 | 0.74 | 0.27 | 0.68 | 0.18 | 0.39 | 0.13 | 0.24 | 0.11 |
| TriSense (7B) | 64 | **1.12** | 0.42 | 0.80 | 0.28 | 0.57 | 0.21 | 0.43 | 0.22 |
| TriSense (7B) | 128 | **1.12** | **0.43** | **0.87** | **0.31** | **0.64** | **0.32** | **0.49** | **0.26** |

**Ablation Studies.** We conduct ablation experiments to assess the contribution of different components, including the **training strategy**, the **Query-Based Connector**, and the **number of frames** processed by the model. As shown in Table 7 and Table 8, we compare our adaptive weighting strategy against simpler alternatives. The **Addition** baseline directly sums the modality features without weighting. The **Fixed Weights** baseline assigns equal weights (e.g., 0.33 each in AVS), or

fixed values depending on the modality pair (e.g., 0.5 for each active modality and 0 for inactive ones in VS/AV), and uses only the visual stream in visual-only tasks (weight of 1, others set to 0). These comparisons confirm the effectiveness of our query-adaptive weighting mechanism.

Table 8: **Ablation results on Segment Captioning.** We analyze the effects of training stages, connector design, and input frame count across four modality settings (AVS, VS, AV, V). Metrics include BLEU-4 (B), METEOR (M), ROUGE-L (R), and CIDEr (C).

| Model | Frame Number | AVS-SC | | | | VS-SC | | | | AV-SC | | | | V-SC | | | |
|---|---|---|---|---|---|---|---|---|---|---|---|---|---|---|---|---|---|
| | | B | M | R | C | B | M | R | C | B | M | R | C | B | M | R | C |
| *Training Stages* | | | | | | | | | | | | | | | | | |
| Stage1 Only | 64 | 0.0 | 1.5 | 4.3 | 0.1 | 0.0 | 1.4 | 4.1 | 0.1 | 0.0 | 1.3 | 4.1 | 0.1 | 0.0 | 1.3 | 4.0 | 0.1 |
| Stage1+2 | 64 | 2.1 | 9.3 | 19.8 | 6.6 | 1.8 | 8.6 | 20.2 | 6.7 | 3.0 | 11.1 | 21.5 | 8.2 | 5.3 | 6.2 | 11.8 | 13.8 |
| *Connector* | | | | | | | | | | | | | | | | | |
| Addition | 64 | 1.6 | 9.9 | 18.0 | 5.8 | 1.8 | 9.1 | 19.3 | 5.8 | 3.8 | 11.2 | 22.0 | 11.5 | 5.7 | 11.1 | 28.5 | 21.9 |
| Fixed Weights | 64 | 3.1 | 9.8 | 19.4 | 6.6 | 2.4 | 9.3 | 20.7 | 7.9 | 4.3 | 11.8 | 26.1 | 15.3 | **7.4** | 12.7 | 30.6 | **36.7** |
| *Frame Number* | | | | | | | | | | | | | | | | | |
| TriSense (7B) | 32 | 3.2 | 9.7 | 19.9 | 7.7 | 2.1 | 9.3 | 19.5 | 7.9 | 3.4 | 11.1 | 22.5 | 9.6 | 6.3 | 11.7 | 29.8 | 29.2 |
| TriSense (7B) | 64 | **3.4** | 10.1 | 20.1 | 8.3 | 3.0 | **10.0** | 22.2 | 11.8 | 5.3 | 12.2 | 26.3 | 15.4 | 7.3 | 12.6 | 30.7 | 36.3 |
| TriSense (7B) | 128 | **3.4** | 10.2 | 20.2 | 8.5 | 3.1 | 9.9 | 22.8 | 11.5 | 5.4 | 12.3 | 26.7 | 15.4 | 7.3 | 12.8 | 30.8 | 36.1 |

We can observe that in *Training Stages* section, Stage 1 focuses solely on modality alignment and no temporal information is included, therefore does not performing well across two tasks. However, after training in Stage 2, the model acquires around 50% of its capability. For the *Connector* ablation, we find that simply adding all modalities together does not allow the model to emphasize the more important modalities, resulting in a drop in performance. Similarly, in the *Fixed Weights* ablation for AVS/VS/AV tasks, assigning equal weights to modalities fails to effectively capture their varying importance, leading to inferior performance compared to using dynamic modality weights. However, fixing visual weight to 1 indeed leads to slightly better performance in visual-only (V) setting. We also observe that increasing the number of frames used by the model (from 64 to 128) leads to performance improvement in every scenario. This trend is consistent with the findings in both of the Moment Retrieval and Segment Captioning.

## 6 Conclusion

In this work, we introduced TriSense, a novel multimodal large language model designed to advance comprehensive video understanding by integrating visual, audio, and speech modalities. At the core of our model is the Query-Based Connector, which enables dynamic, modality-adaptive fusion—allowing the system to operate effectively across arbitrary combinations of input modalities. This capability is essential for real-world scenarios, where certain modalities may be partially available or entirely absent. To support progress in this area, we constructed TriSense-2M, a large-scale dataset containing over 2 million carefully curated samples. These samples span diverse scenes, durations, and modality alignments, offering a rich foundation for training and evaluation. Through extensive experimentation, we demonstrated that TriSense consistently achieves state-of-the-art performance on key temporal video understanding tasks, including moment retrieval and long-form video segment captioning. Our modality-adaptive framework marks a substantial step toward more flexible and human-like video understanding systems. It not only delivers strong performance in controlled evaluations but also shows robust applicability in real-world conditions with varying input configurations. We believe that both TriSense and the TriSense-2M dataset will serve as valuable resources for future research in multimodal learning and temporal reasoning, enabling broader advances across a range of video understanding applications.

## 7 Acknowledgements

This project is jointly supported by the University of Western Australia (UWA) HDR Scholarship, Australian Research Council ARC DP210101682, DP210102674, and the Australian Government through the Australian Research Council's DECRA funding scheme DE250100030. We thank Zhejiang Lab for providing the computational resources.

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

# Contents of Appendix

# A   Implementation Details

## A.1   Training Recipe

Our training process follows a structured **three-stage** approach: Feature Alignment, Connector Generalization, and Instruction Tuning—to progressively equip the model with strong multimodal and temporal reasoning capabilities. The specific components trained at each stage are illustrated in Figure 4.

**Feature Alignment.** In the first stage, only the Query-Based Connector and LM Head are set as trainable, while all other components remain frozen. This phase employs single-modality inputs, enabling the model to gain an initial understanding of each of the three modalities individually. It also helps the model learn to assign weights effectively in the absence of multimodal context, promoting a focused grasp of modality-specific features without interference from other components.

**Connector Generalization.** During the second stage, we incorporate mixed-modality data and allow training for the Query-Based Connector, Time Encoder, Time Head, and LM Head, while keeping the LLM backbone fixed. This phase equips the connector to handle weight allocation across multiple modalities, thereby enhancing its generalization beyond isolated modalities. Simultaneously, training the Time Encoder and Time Head introduces the model to temporal structure, laying the groundwork for capturing inter-modality dynamics over time.

**Instruction Tuning.** In the final stage, we freeze the Query-Based Connector and train the remaining components—including the Time Encoder, Time Head, LM Head, and the LLM backbone—using mixed-modality inputs. By keeping the connector fixed, we retain the modality alignment learned in previous stages. This step concentrates on refining temporal reasoning and language understanding capabilities, strengthening the LLM's ability to interpret and process multimodal, temporally-sensitive queries across diverse scenarios.

## A.2   Datasets

In alignment with the objectives of the three-stage training process outlined earlier, we employ varying datasets and data volumes at each stage. The overarching goal is to enhance the model's capacity for temporal video understanding while retaining robust general video understanding abilities. An overview of the datasets used in each stage is provided in Table 9.

Table 9: Datasets and sample sizes used across the three training stages.

| Stages | Datasets | Total Quantity |
|---|---|---|
| Stage 1 | Clotho [6], LLaVA-LCS558K [21], Valentini-Botinhao Speech Dataset [29] | 600K |
| Stage 2 | TriSense-2M (880K), LLaVA-Video-178K (120K) [37] | 1M |
| Stage 3 | TriSense-2M (1.12M), LLaVA-Video-178K (380K) [37] | 1.5M |

**Stage1**. For the initial stage, we use a combination of the **Clotho** [6], **LLaVA-LCS558K** [21], and **Valentini-Botinhao Speech Dataset** [29] as training dataset:

- **Clotho** is an audio captioning dataset containing 4,981 audio clips, each paired with five unique captions, totaling 24,905 annotations. The audio clips range from 15 to 30 seconds, and each caption consists of 8 to 20 words.

- **LLaVA-LCS558K** is a concept-balanced multimodal dataset comprising 558,000 image-text pairs, annotated using BLIP-generated captions. It is designed to support feature alignment during the pretraining of vision-language models.

- **Valentini-Botinhao Speech Dataset** is a parallel corpus of clean and noisy speech recordings. It is widely used in training and evaluating speech enhancement and text-to-speech (TTS) systems, featuring 48kHz audio from multiple speakers under various noise conditions.

**Stage 2 and 3**. For these stages, we adopt our newly proposed TriSense-2M dataset, applying a 9:1 training-to-testing split. This results in 1.9 million training samples and 0.1 million test samples. The training data is further partitioned into approximately 880K samples for Stage 2 and 1.12M samples for Stage 3.

To ensure the model also retains general video understanding capabilities, we supplement the training data with a portion of LLaVA-Video-178K [37], which includes video captioning, open-ended QA, and multiple-choice QA tasks. This mixed-task dataset helps the model develop broader understanding skills beyond temporal reasoning.

Table 10: Details of test set.

| Num. Videos | Num. Events | Avg. Event Duration | Total Tasks |
|:---:|:---:|:---:|:---:|
| 3805 | 11415 | 7.23s | 91320 |

To avoid massive evaluation time, we extract 11,415 challenging samples from the 0.1M test set, as shown in Table 10 using two filtering criteria: **1)** The majority of events should occur in the middle portion of the video rather than the beginning. **2)** Captions must contain at least 20 words. Evaluation is conducted using a single A100 SXM4 80GB GPU with a batch size of 1, requiring approximately 8–10 hours to complete. All models in comparison are evaluated on this same test subset, using their officially recommended hyperparameters (e.g., number of frames, temperature, top-p, etc.)

### A.3 Detailed training settings

Our multimodal framework incorporates dedicated encoders for each modality. For the visual modality, we adopt *openai/clip-vit-large-patch14-336* [25]; for audio and speech modalities, we employ *BEATs_iter3+ (AS2M) (cpt2)* [4] and *Whisper-large-V3* [26], respectively. As for the large language model (LLM) backbone, we select Mistral-7B [17], initialized from TRACE [11], instead of using other LLM backbones, rather than using other LLM backbones. This choice is motivated by TRACE's prior training on large-scale temporal understanding data, which equips it with stronger temporal reasoning capabilities. The maximum context length is configured to 4096 tokens.

During training, videos are resampled at 1 frame per second (fps) to improve efficiency—this step is omitted during inference to retain full fidelity. This resampling reduces input redundancy and accelerates training.

In Stage 1, we train the model with a batch size of 512 and single-frame input, completing within 10 hours using 4×A100SXM4-80GB GPUs. Stage 2 and 3 are conducted on 16×A100SXM4-80GB GPUs with batch sizes of 128 and 256, respecitvely. Stage 2 requires approximately 3.5 days to complete, and stage 3 requires 7 days to finish. We use DeepSpeed Zero2 because the BEATs model does not function properly under Zero3 settings. Further details on datasets and hyperparameters are provided in 11.

Table 11: Training configurations and hyperparameters by stage.

| Settings | Stage 1 | Stage 2 & Stage3 |
|---|---|---|
| Computation | 4×A100SXM4-80GB. | 16×A100SXM4-80GB |
| Vision Encoder | clip-vit-large-patch14-336 | clip-vit-large-patch14-336 |
| Audio Encoder | BEATs_iter3+ (AS2M) (cpt2) | BEATs_iter3+ (AS2M) (cpt2) |
| Speech Encoder | Whisper-large-V3 | Whisper-large-V3 |
| DeepSpeed Stage | Zero2 Offload | Zero2 Offload |
| LLM Backbone | Mistral-7B-v0.2 | Mistral-7B-v0.2 |
| Batch Size | 512 | 128 & 256 |
| Num Frames | 1 | 64 |
| Frame Sample | Uniform | Uniform |
| Train Epochs | 2 | 1 |
| Learning Rate | 1e-3 | 5e-6 |
| LR Scheduler | Cosine | Cosine |
| Model Max Length | 4096 | 4096 |
| Training Duration | 10 Hours | 3.5 days & 5.5 days |

## B   More details of TriSense-2M

### B.1   Training Generator and Judger

This section describes the data generation and manual filtering process used to prepare training data for both the **Generator** and the **Judger**. To ensure efficiency and quality in omni-modal caption generation, we first utilize GPTo1 [16] to produce high-quality annotation samples for Supervised Fine-tuning (SFT). The specific prompts used for this process are shown in 5. These prompts serve a dual purpose: they are used to generate training data via GPT and also function as system prompts during the SFT of both the Generator and the Judger.

To further enhance caption quality, we implement a two-stage scoring mechanism. After captions are generated, GPT conducts a self-evaluation. Then, a separate GPT instance provides an additional evaluation to filter out low-quality samples. Following this automated scoring, we conduct manual sampling to verify consistency and ensure a high quality standard is met.

Data is generated in batches of 1,000 samples. From each batch, we randomly select 500 samples for manual review to evaluate the generated content and the reliability of GPT's scoring. If over 80% of the reviewed samples meet our quality criteria, the batch is retained; otherwise, it is discarded.

Ultimately, we curate 10,000 training samples for the Generator and 3,000 samples for the Judger. Both models are trained for 3 epochs to establish effective captioning and judging capabilities.

### B.2   Training data format

This section outlines the data format used for training TriSense. We adopt a ShareGPT-style format, where each training sample consists of 8 conversation rounds, each corresponding to a different modality combination. The tasks and settings within these rounds are randomized—for instance, one round might involve a VS-SC task, while the next could be an AVS-SC or V-MR task.

Following the approach in TRACE, we use special tokens such as ⟨sync⟩ and ⟨time⟩ to signal the model to switch between different prediction heads. An example of the data structure is illustrated in 6.

## C   Additional Experiments.

### C.1   More ablation studies

We conduct small-scale ablation studies on slot compression and task-specific heads. Specifically, we randomly selected 20K samples and trained for 3 epochs, initializing all weights from [11]. All experiments were conducted with a sampled frame rate of 64. In terms of token compression, we explored different slot configurations, including 8, 32, and 64. For the prediction head, we removed

Input Examples:
Audio: {audio\_caption}
Visual: {visual\_caption}
Speech: {speech\_caption}

You should generate captions according to the following rules:
1. The generated captions must preserve all essential information from the original captions.
2. Do not introduce any content that is not present in the original captions.
3. Do not copy verbatim from the original captions; instead, paraphrase the key information and incorporate it into the new caption.
4. Each generated caption must not exceed 200 words.

Please think carefully and provide your answer in JSON format as follows: { "AVS": "", "AV": "", "VS": "", "Score": ""}. Note that only one caption should be provided for each of AVS, AV, and VS. You MUST respond in JSON format.

Input Examples:
Audio: {audio_caption}
Visual: {visual_caption}
Speech: {speech_caption}
AVS: {audio-visual-speech_caption}
VS: {visual-speech_caption}
AV: {audio-visual_caption}

Assign a score from 1 to 5. If the quality is very poor, assign a score of 1. If the generated captions satisfy approximately 80% of the above criteria, assign a score of 3. If they fully satisfy all criteria (100%), assign a score of 5.

Please analyze carefully and provide your evaluation in the following JSON format: { "AVS": "", "AV": "", "VS": "", "Score": "" }Note that you should provide only one score and one caption for each of AVS, AV, and VS. You MUST respond in JSON format.

Figure 5: **Prompts used for training the Generator and Judger.** The left prompt guides GPT in generating omni-modal captions for the Generator using audio, visual, and speech inputs. The right prompt is used to train the Judger by instructing GPT to assess the quality of generated captions based on coverage, accuracy, and paraphrasing. During data creation, samples are randomly selected and manually filtered to ensure high-quality training data.

the Time Head and instead encoded temporal information into special tokens, which were injected into the LLM. Prediction was then performed using only the LM Head.

Table 12: Ablation study on the slot compression [12].

| AVS-SC | AVS-MR | Avg. Event Duration |
|---|---|---|
| Slot=8 | B: 1.6 M: 3.1 R: 13.1 C: 1.9 | IoU=0.5: 0.3 IoU=0.7: 0.1 |
| Slot=16 | B: 1.8 M: 3.4 R: 14.6 C: 2.4 | IoU=0.5: 0.4 IoU=0.7: 0.3 |
| Slot=32 | B: 1.8 M: 3.4 R: 15.6 C: 2.4 | IoU=0.5: 0.4 IoU=0.7: 0.3 |
| Slot=64 | B: 1.9 M: 3.5 R: 16.1 C: 2.4 | IoU=0.5: 0.5 IoU=0.7: 0.2 |

Table 13: Ablation study on the task-specific heads.

| | AVS-SC | AVS-MR |
|---|---|---|
| Unified Head | B: 1.5 M: 3.1 R: 13.2 C: 1.1 | IoU=0.5: 0.2 IoU=0.7: 0.1 |
| Time Head + LM Head | B: 1.8 M: 3.4 R: 14.6 C: 2.4 | IoU=0.5: 0.4 IoU=0.7: 0.3 |

As shown in Table 12, for slot compression, although reducing the compression rate (by increasing the number of slots) can bring slight performance improvements, it also results in a significant increase in computational overhead. Therefore, the benefits are subject to diminishing returns. As shown in the table, setting the slot to 16 currently offers a good balance between performance and computational cost. For task-specific head in Table 13, using a single unified head alone leads to a significant drop in performance, indicating that the time head provides additional temporal information which is beneficial to the model. There may be better unified solutions in the future.

We also provide more ablation studies in Table 14 and Table 15 for different branches of the Query-Based Connector based on the same 20K randomly selected samples as above. For example, AV-Only in the table means we discarded the Speech branch. During training, all the weigths are initialized from [11], and LLM backbone is set to frozen. Results show that all three modalities contribute positively to all tasks, with the full AVS configuration performing best.

```
Dataset Format Example of ShareGPT

{
  "video": "zEqicUE2R0I.mp4",
  "conversations":
  [
    {
      "from": "human",
      "value": "<video>\nFrom 1070.8 to 1081.9, please detail the visual-sound event, highlighting any essential visuals and
sounds."
    },
    {
      "from": "gpt",
      "value": "<sync><time><time><time><time><time><time><time><time><time><time><time><time><time><time>A man is
passionately speaking on stage while the audience laughs and talks in the theater. The crowd is engaged and responsive."
    },
    {
      "from": "human",
      "value": "Could you provide a summary of the video between [1070.8, 1081.9], focusing on visual, audio, and speech
elements?"
    },
    {
      "from": "gpt",
      "value": "<sync><time><time><time><time><time><time><time><time><time><time><time><time><time><time>A
person is performing on stage, and the audience is laughing and talking in a theater, but the speech subtitle mentions that
the person has to hide because they look like a rat from the movie Ratatouille."
    },
    {
      "from": "human",
      "value": "<sync><time><time><time><time><time><time><time><time><time><time><time><time><time><time>Please
find the precise time in seconds when the content described as 'a man is singing on stage' takes place in the video, and
report this timestamp."
    },
    {
      "from": "gpt",
      "value": "<sync><time><time><time><time><time><time><time><time><time><time><time><time><time><time>a man is
singing on stage"
    }
    <Remaining conversations......>
  ]
  "times": [[1070.8, 1081.9], [1070.8, 1081.9], [1070.8, 1081.9], <remaining times......>]
}
```

Figure 6: **Example of ShareGPT-style annotation format used during training.** Each sample includes multi-turn conversations over video segments with synchronized modality cues. Only the first three rounds are shown due to space limitations.

Table 14: Ablation studies on the branches of Query-based Connector on Segment Captioning tasks.

| Model | AVS-SC | VS-SC | AV-SC | V-SC |
|---|---|---|---|---|
| V-Only | B: 1.0 M: 1.7 R: 10.1 C: 0.9 | B: 1.1 M: 1.6 R: 10.3 C: 1.1 | B: 1.0 M: 1.7 R: 10.2 C: 1.1 | B: 1.0 M: 1.8 R: 13.3 C: 1.4 |
| VS-Only | B: 1.7 M: 2.9 R: 13.3 C: 2.4 | B: 1.6 M: 2.9 R: 14.7 C: 2.5 | B: 0.7 M: 2.3 R: 13.2 C: 1.5 | B: 0.7 M: 1.6 R: 13.2 C: 1.2 |
| AV-Only | B: 1.7 M: 3.1 R: 13.7 C: 1.9 | B: 1.4 M: 2.2 R: 12.0 C: 2.1 | B: 0.9 M: 3.1 R: 15.0 C: 1.7 | B: 0.7 M: 1.6 R: 13.0 C: 1.1 |
| AVS-Full | B: 1.8 M: 3.4 R: 14.6 C: 2.4 | B: 1.7 M: 3.1 R: 15.1 C: 2.5 | B: 1.0 M: 3.2 R: 15.3 C: 1.7 | B: 0.8 M: 1.8 R: 13.4 C: 1.5 |

## C.2 General understanding dataset

For general video understanding evaluation, we report results on VideoMME [7]—a large-scale benchmark designed to assess multimodal large language models (MLLMs) in video analysis. VideoMME covers a wide range of visual domains, temporal scales, and modalities, including 900 videos (totaling 254 hours) and 2,700 human-annotated QA pairs. As shown in Table 16, our model not only demonstrates significant advantages in multimodal scenarios but also performs competitively in general understanding tasks. It is worth noting that our model uses **much less** general understanding data (500K) compared to TRACE-uni [11], which uses 0.9M, as reported in their official paper.

Table 15: Ablation studies on the branches of Query-based Connector on Moment Retrieval tasks.

| Model | AVS-MR | VS-MR | AV-MR | V-MR |
|---|---|---|---|---|
| V-Only | IoU=0.5: 0.1 IoU=0.7: 0.0 | IoU=0.5: 0.1 IoU=0.7: 0.0 | IoU=0.5: 0.1 IoU=0.7: 0.1 | IoU=0.5: 0.4 IoU=0.7: 0.2 |
| VS-Only | IoU=0.5: 0.4 IoU=0.7: 0.1 | IoU=0.5: 0.3 IoU=0.7: 0.2 | IoU=0.5: 0.1 IoU=0.7: 0.0 | IoU=0.5: 0.3 IoU=0.7: 0.1 |
| AV-Only | IoU=0.5: 0.4 IoU=0.7: 0.2 | IoU=0.5: 0.1 IoU=0.7: 0.1 | IoU=0.5: 0.3 IoU=0.7: 0.1 | IoU=0.5: 0.3 IoU=0.7: 0.1 |
| AVS-Full | IoU=0.5: 0.4 IoU=0.7: 0.3 | IoU=0.5: 0.3 IoU=0.7: 0.2 | IoU=0.5: 0.3 IoU=0.7: 0.2 | IoU=0.5: 0.4 IoU=0.7: 0.2 |

Table 16: Zero-shot performance on general understanding dataset Video-MME [7]. Our model only uses **55%** of the general understanding data compared to TRACE-uni.

| Model | VideoMME (Overall Scores w/o Subtitles) |
|---|---|
| VideoChat2 (7B) | 33.7 |
| Video-LLaVA (7B) | 39.9 |
| VideoLLaMA2 (7B) | 46.6 |
| TRACE (7B) | 43.8 |
| TRACE-uni (7B) | **49.6** |
| TriSense (7B) | 48.7 |

# D   Case studies on diferent scenarios

We provide case studies of TriSense in various scenarios in Figures 7 to 12.

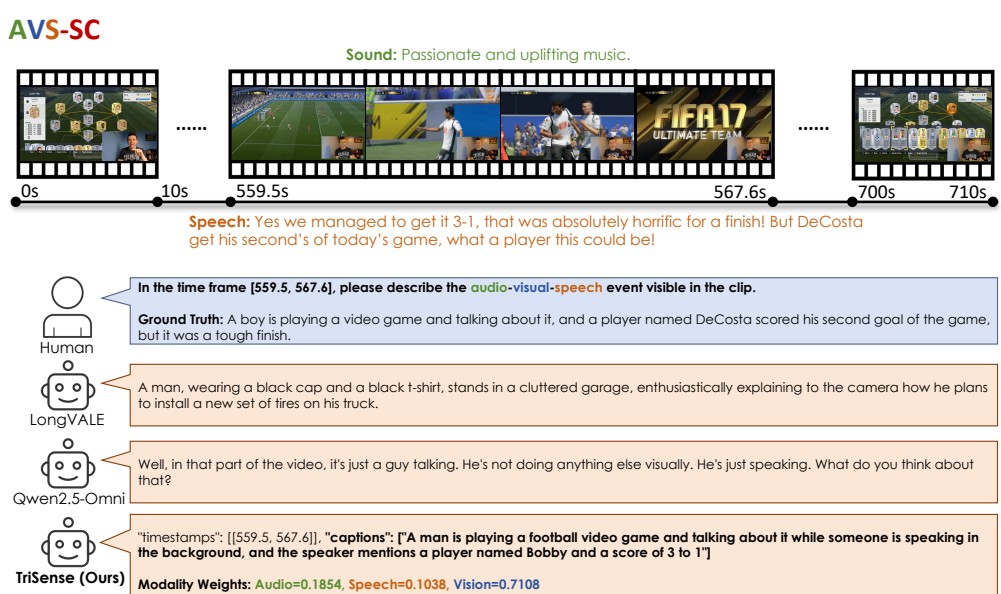

Figure 7: Case study of TriSense on AVS-SC task.

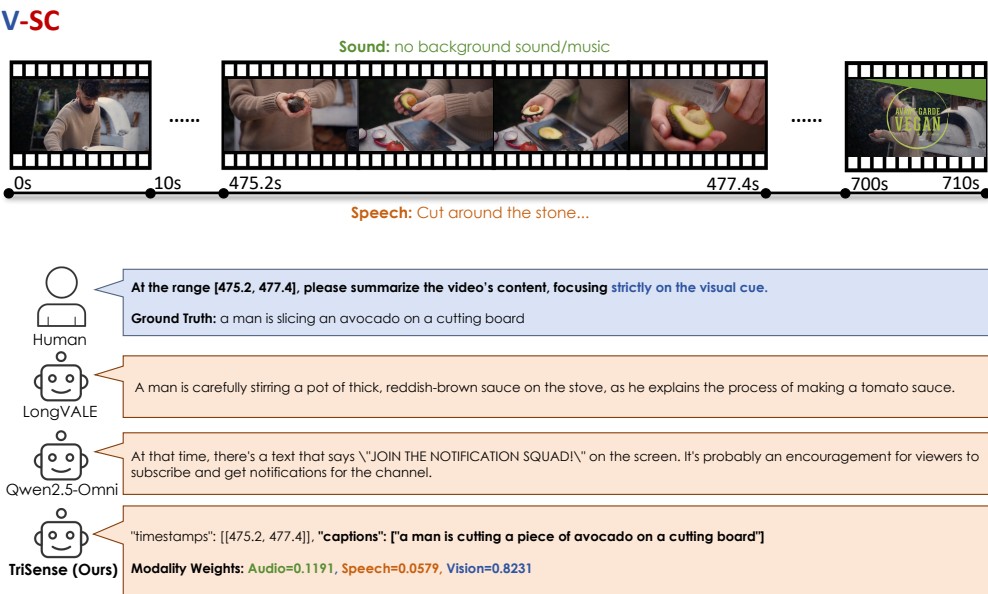

Figure 8: Case study of TriSense on V-SC task.

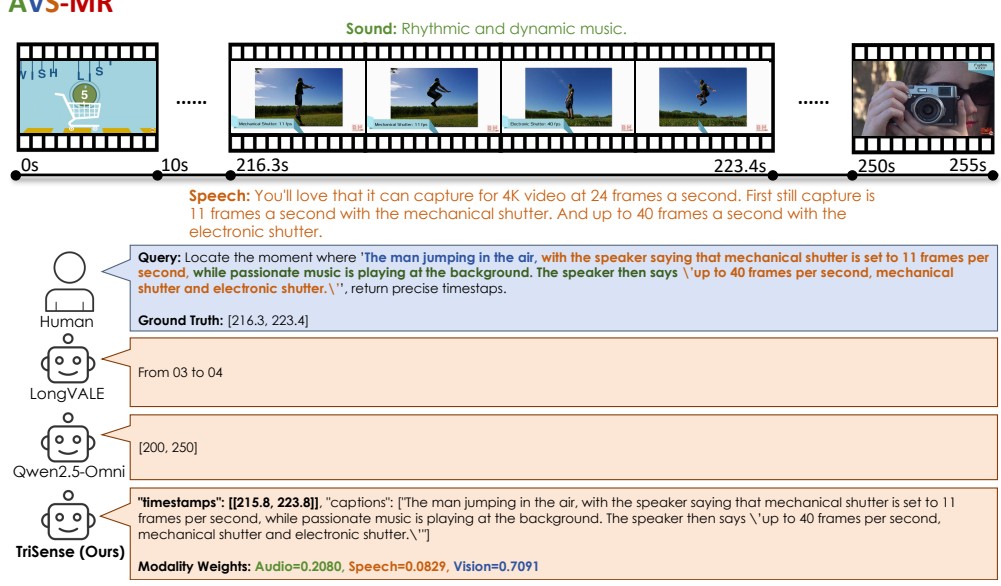

Figure 9: Case study of TriSense on AVS-MR task.

## VS-MR

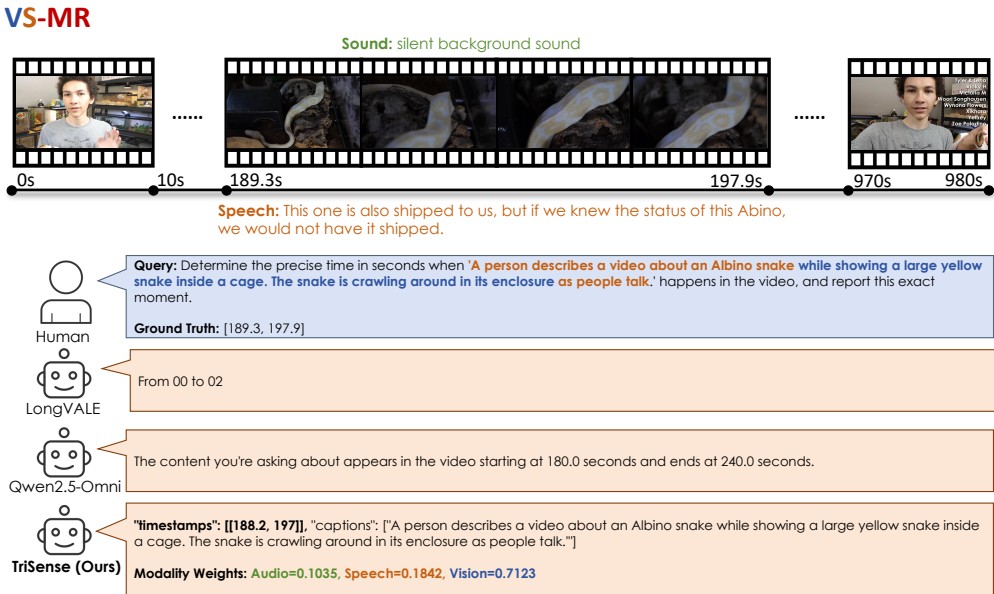

Figure 10: Case study of TriSense on VS-MR task.

## AV-MR

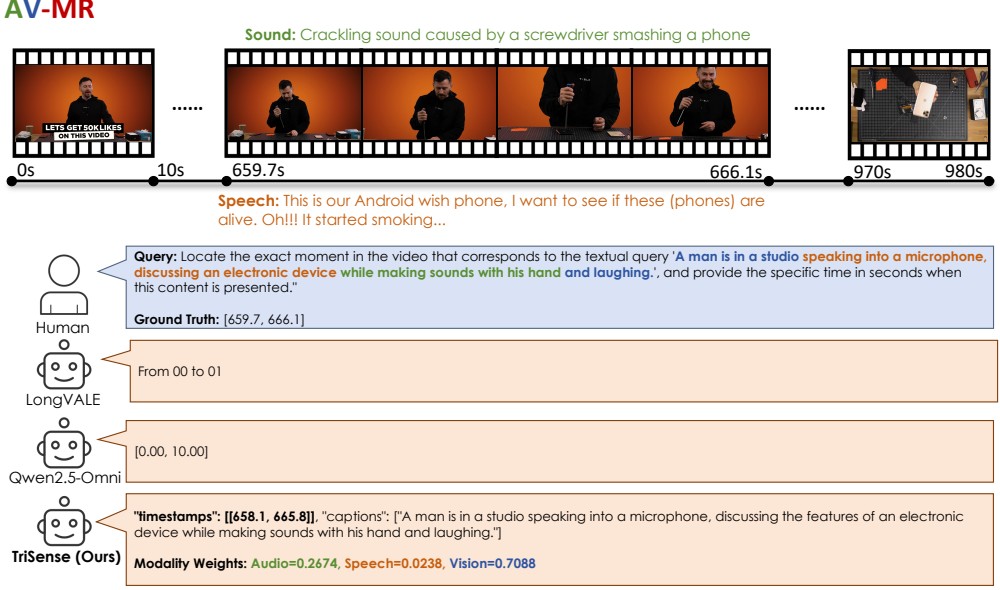

Figure 11: Case study of TriSense on AV-MR task.

## General Understanding

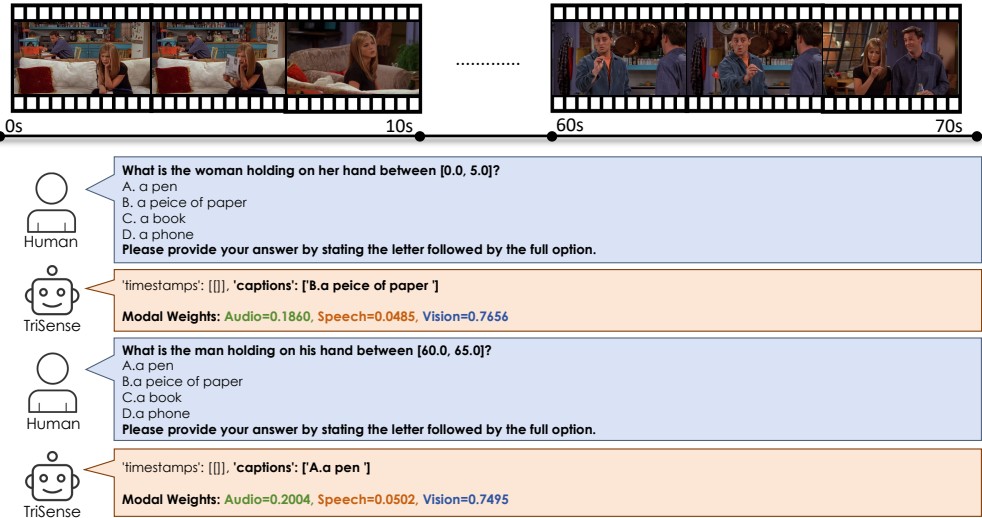

Figure 12: Case study of TriSense on General Understanding task.

