# OpenReview forum: "Watch and Listen: Understanding Audio-Visual-Speech Moments with Multimodal LLM"
_NeurIPS.cc/2025/Conference — NeurIPS 2025 poster_

### Official Review · Reviewer_9tN8 · 2025-06-30

**Clarity:** 3
**Significance:** 2
**Originality:** 3
**Rating:** 4
**Confidence:** 4

**Summary:**

The authors design a video understanding model integrating audio, speech and visual modalities, build a three-modal caption dataset for training, and achieved promising results in temporal video understanding tasks, e.g., Moment Retrieval.

**Questions:**

* Why do the authors only focus on temporal video understanding tasks rather than more general video understanding benchmarks (e.g., VideoMME)?
* The design of query-based connector to accommodate multi-modal inputs has been explored in prior work (e.g., Video-LLaMA). Is it appropriate to claim this as a core technical contribution at the model architecture level?
* The authors aim to enhance video understanding (especially temporal tasks like Moment Retrieval and Segment Caption) by integrating audio-visual information, but lack ablation studies to demonstrate whether Speech/Audio modalities truly improve task performance.

**Ethical Concerns:**

["Major Concern: Improper research involving human subjects", "Major Concern: Data privacy, copyright, and consent"]

**Final Justification:**

The authors have addressed my questions comprehensively and clearly, and I have basically no further doubts about this paper. However, the dataset proposed by the authors does not consider verifying whether it can bring about the improvement of General Video Understanding capabilities, which may limit the extensibility of this work.

**Limitations:**

Yes

**Quality:**

2

**Strengths And Weaknesses:**

### Strengths
* Achieve promising results on temporal video understanding benchmarks.
* Make substantial data contributions, which are significant for omni-model exploration.

### Weaknesses

* The authors focus only on simple and basic benchmarks/tasks, without considering implications for general video or audio-visual understanding.
* The authors claim existing models fail to fuse audio information but doesn’t explain why. The proposed architecture resembles prior designs, suggesting the key contribution lies in the dataset.
* The authors introduce audio/speech modalities without ablation studies or case demonstrations to validate their impact on temporal video understanding.

---

> ### Author Rebuttal · Authors · 2025-07-30
>
> **First of all, we would like to sincerely thank the four reviewers for their constructive feedback on our manuscript:** "The paper is clearly written and easy to follow, with well-organized sections and clear explanations of both the method and experiments", "The introduction of the new TriSense 2M dataset is a valuable contribution to the community" (Reviewer ThZF); "This paper studies a very interesting problem which seeks to build an omni-modal large language model for comprehensive multi-modal understanding" (Reviewer 1DNL); "The authors propose a groundbreaking adaptive weighting framework validated by TriSense-2M" (Reviewer NaTe); "Make substantial data contributions, which are significant for omni-model exploration" (Reviewer 9tN8). **Next, we will carefully address each of the reviewers’ comments.**
>
> > Q1: The authors focus only on simple and basic benchmarks/tasks, without considering implications for general video or audio-visual understanding.
>
> We would like to clarify that our task is **NOT simple**, we considered not only **audio-visual-speech tasks** and all their combinations (including audio-visual understanding), but also **general understanding** tasks (Table 10).
>
> Firstly, our proposed task requires the model to **natively support full-modality input** and perform temporal omni-modal captioning and moment retrieval, while also supporting **modality dropout**—features that most VLMs do not support. Secondly, as shown in Figure 3, we provide a detailed breakdown of video durations in TriSense-2M, where **83.5% of the videos are long-form (10–20 minutes)**, demanding accurate temporal understanding in long-form challenging settings. Finally, as shown by our experiments in Section 5, even models like Qwen2.5-Omni, which benefit from higher-quality training data and more training resources, struggle to perform well on our proposed task. This further demonstrates that our task presents a meaningful challenge for existing VLMs.
>
> > Q4: Why do the authors only focus on temporal video understanding tasks rather than more general video understanding benchmarks?
>
> We would like to kindly emphasize that the dataset and model proposed in this paper are specifically designed to address the poor performance of current general VLMs in **omni-modal temporal understanding** scenarios. It is the temporal understanding that we aim to address, rather than general video understanding tasks. This is why we chose to focus on this direction.
>
> Additionally, we have shown that the TriSense model not only outperforms existing VLMs in temporal understanding, but also **remains competitive in general understanding tasks**, even when trained with only **55%** of the general data, as we have included the **VideoMME** results in the Appendix C for your reference. We plan to include more general data in the future for a stonger model with both solid temporal & general understaning abilities.
>
> > Q2 & Q5: The authors argue that existing models struggle with audio fusion but do not explain why. Since similar query-based connector designs have appeared in prior work, is it appropriate to consider this a core technical contribution, or is the main novelty in the dataset?
>
> Analysis was added in **Sections 1 (challenges) and 2 (last paragraph)** of the original paper, and more will be added in the supplementary material. Besides, we would like to respectfully claim that the proposed Query Based Connector **has NOT** been used by Video-LLaMA and VideoLLaMA 2, **especially for the modality reweighting mechanism,** with detailed explanations below:
>
> - **Architectural differences:**
>   1. **Video-LLaMA [1]** uses QFormer [2], which levergaes **fixed, learnable query vectors** to perform cross-attention across modalities, generating instruction-aware, fixed-length representations. In contrast, our method uses the **actual user-input prompt** as the query, making it flexible and only used in the **initial stage.** Because the cross-attention facilitates cross-modal interaction, placing the query alongside the target modality is a **natural and intuitive design choice.** Therefore, despite similarities, our approach **differs significantly** from QFormer in both the nature of the query and our overall objectives, as cross-attention is not the central contribution of our work.
>   2. **VideoLLaMA 2 [3]:** Although VideoLLaMA 2 also incoporates audio stream, the audio connector is just a simple MLP to project the audio feature, which is totally different from the Query-Based Connector proposed in our paper.
>   3. **Query-Based Connector:** The query attends to each modality (visual, audio, speech) separately via cross-attention to obtain query-relevant features (note: although this is similar to QFormer, it’s only our initial step). Then, all modalities pass through an MLP and softmax to obtain **dynamic weights** for each modality according to the query, enabling **modality reweighting** and temporal-aligned fusion that naturally masks out missing modalities. **Note that both of the Video-LLaMA or VideoLLaMA 2 have no modality reweighting mechanism.**
>
> - **Design Intentions Difference:** The primary purpose of QFormer is to extract single-modality features in order to **bridge the modality gap**. The self-gated Multimodal Query Fusion proposed by CREMA [4] is designed to **prevent the query token size from growing linearly with each new modality.** In contrast, our Query-Based Connector is designed to provide a **universal, simple, and effective way to extract features from all modalities and dynamically weight and fuse them**. This design philosophy naturally favors a simple connector structure, that is, making it too complex would risk reduced generalizability, overfitting, and optimization difficulties.
>
> Lastly, we want to emphasize that our key contribution lies in demonstrating that such a concise and adaptable architecture can perform well across diverse modalities, which **has not been shown in prior work.** We believe this offers meaningful insights for future research. The proposed TriSense-2M dataset is intended to serve as a **robust pretraining foundation** for large multimodal models. Additionally, the TriSense model trained on this dataset can act as a **strong foundation model** for future studies, supporting fine-tuning and enhancement while substantially reducing training costs. Therefore, while our dataset is a key contribution, the Query-Based Connector is also a **core technical contribution**.
>
> > Q3 & Q6: The authors introduce audio and speech modalities to enhance temporal video understanding, but provide no ablation studies or case examples to show whether these modalities actually improve performance.
>
> We provide the following ablation study **trained from scratch**. We selected 20K samples and trained for 3 epochs, removing certain branches of the Query-Based Connector. For example, **AV-Only** in the table means we discarded the Speech branch. During trainning, all the weigths are initialized from [4], and LLM backbone is set to frozen. Results show that all three modalities **contribute positively** to all tasks, with the full AVS configuration performing best. Due to resource and time constraints, we are unable to conduct larger-scale experiments at this time, but we plan to include them in future versions of the paper.
>
> | Model    | AVS-SC                                       | VS-SC                                        | AV-SC                                        | V-SC                                         |
> | -------- | -------------------------------------------- | -------------------------------------------- | -------------------------------------------- | -------------------------------------------- |
> | V-Only   | **B:** 1.0 **M:** 1.7 **R:** 10.1 **C:** 0.9 | **B:** 1.1 **M:** 1.6 **R:** 10.3 **C:** 1.1 | **B:** 1.0 **M:** 1.7 **R:** 10.2 **C:** 1.1 | **B: 1.0 M: 1.8 R:** 13.3 **C:** 1.4         |
> | VS-Only  | **B:** 1.7 **M:** 2.9 **R:** 13.3 **C: 2.4** | **B:** 1.6 **M:** 2.9 **R:** 14.7 **C: 2.5** | **B:** 0.7 **M:** 2.3 **R:** 13.2 **C:** 1.5 | **B:** 0.7 **M:** 1.6 **R:** 13.2 **C:** 1.2 |
> | AV-Only  | **B:** 1.7 **M:** 3.1 **R:** 13.7 **C:** 1.9 | **B:** 1.4 **M:** 2.2 **R:** 12.0 **C:** 2.1 | **B:** 0.9 **M:** 3.1 **R:** 15.0 **C: 1.7** | **B:** 0.7 **M:** 1.6 **R:** 13.0 **C:** 1.1 |
> | AVS-Full | **B: 1.8 M: 3.4 R: 14.6 C: 2.4**             | **B: 1.7 M: 3.1 R: 15.1 C: 2.5**             | **B: 1.0 M: 3.2 R: 15.3 C: 1.7**             | **B:** 0.8 **M: 1.8 R: 13.4 C: 1.5**         |
>
> | Model    | AVS-MR                        | VS-MR                         | AV-MR                         | V-MR                          |
> | -------- | ----------------------------- | ----------------------------- | ----------------------------- | ----------------------------- |
> | V-Only   | **R@0.5:** 0.1 **R@0.7:** 0.0 | **R@0.5:** 0.1 **R@0.7:** 0.0 | **R@0.5:** 0.1 **R@0.7:** 0.1 | **R@0.5: 0.4 R@0.7: 0.2**     |
> | VS-Only  | **R@0.5:** 0.4 **R@0.7:** 0.1 | **R@0.5: 0.3 R@0.7: 0.2**     | **R@0.5:** 0.1 **R@0.7:** 0.0 | **R@0.5:** 0.3 **R@0.7:** 0.1 |
> | AV-Only  | **R@0.5:** 0.4 **R@0.7:** 0.2 | **R@0.5:** 0.1 **R@0.7:** 0.1 | **R@0.5: 0.3** **R@0.7:** 0.1 | **R@0.5:** 0.3 **R@0.7:** 0.1 |
> | AVS-Full | **R@0.5: 0.4 R@0.7: 0.3**     | **R@0.5: 0.3 R@0.7: 0.2**     | **R@0.5: 0.3 R@0.7: 0.2**     | **R@0.5: 0.4 R@0.7: 0.2**     |
>
> **References:**
>
> [1] Zhang, Hang, et al. "Video-LLaMA: An Instruction-tuned Audio-Visual Language Model for Video Understanding." EMNLP, 2023.
>
> [2] Li, Junnan, et al. "Blip-2: Bootstrapping language-image pre-training with frozen image encoders and large language models." ICLR, 2023.
>
> [3] Cheng, Zesen, et al. "VideoLLaMA 2: Advancing spatial-temporal modeling and audio understanding in video-llms." arXiv, 2024.
>
> [4] Guo, Yongxin, et al. "TRACE: Temporal Grounding Video LLM via Causal Event Modeling." ICLR, 2025.

---

> > ### Comment · Reviewer_9tN8 · 2025-08-07
> >
> > I must admit that there were some oversights in my reading of the authors' work. However, the authors have largely addressed my concerns comprehensively:
> >
> > 1. The authors explained that they do not only focus on basic grounding tasks, but also include segment captioning tasks corresponding to grounding. Results of some general benchmarks, such as VideoMME, are presented in the supplementary materials.
> > 2. The authors provided a reasonable explanation for why existing models struggle with audio fusion.
> > 3. They clearly elaborated that the application of the query-based connector to the reweight mechanism is indeed an original contribution of this paper, which has been verified in Table 5.
> > 4. The ablation experiment conducted by the authors has well validated the significance of introducing audio and speech modalities.
> >
> > Therefore, I am inclined to give a positive score.
> >
> > In addition, I hope the authors can include more benchmarks, such as MVBench and Egoschema. Meanwhile, it is also recommended that the contents mentioned in points 3 and 4 above be incorporated into the main text of the paper.

---

> ### Comment · Area_Chair_698r · 2025-08-05
> **post-rebuttal comments**
>
> Dear reviewer 9tN8,
>
> As you may have seen already, the authors have responded to the questions in your initial review. Can you please share your thoughts post rebuttal, once you've had a chance to see the author responses and also the other reviews?
>
> This is critical in arriving at an informed decision.
>
> Best, AC

---

> ### Author Response · Authors · 2025-08-07
>
> Dear Reviewer 9tN8,
>
> We appreciate your thoughtful feedback and have done our best to address all of your concerns thoroughly. If there’s anything else we can clarify or expand on, we’d be happy to do so.
>
> Since **only two days remain**, we look forward to your response at your early convenience.
>
> Best regards,
> The Authors

---

> ### Author Response · Authors · 2025-08-09
>
> Dear reviewer 9tN8,
>
> Thank you for improving your score and recognizing the value of our paper!
>
> Since we are unable to provide more general understanding experiments in such short time, we will incorporate these experiments in the final version to further strengthen the paper and improve its overall quality. Furthermore, the clarification and ablation study in poin 3 and point 4 will also be included in the final version.
>
> We sincerely appreciate your efforts, your valuable suggestions have helped up improve the quality of our paper.

---

### Official Review · Reviewer_NaTe · 2025-07-02

**Clarity:** 3
**Significance:** 3
**Originality:** 3
**Rating:** 5
**Confidence:** 3

**Summary:**

This paper presents TriSense, a novel triple-modality large language model with a dynamically adaptive Query-Based Connector that enables robust video temporal understanding across arbitrary modality combinations, supported by the TriSense-2M dataset, achieving state-of-the-art performance in segment captioning and moment retrieval under diverse modality dropout scenarios.

**Questions:**

Typo: In table 4, the second-best results are not underlined in ActivityNet-Caption.

1.	In table 4, is the experiment only related to the number of frames? Can you give the results of other quantitative experiments such as inference speed with a generic scenario dataset?
2.	In Section 3, TriSense-2M uses GPT-o1 for Judger training, but GPT-o1's closed-source nature limits reproducibility. Do you have tried to use open-source LLMs as an alternative? Can you give the effect of using different LLMs on the data?
3.	In Section 3, have you calculated the percentage of missing modal data, and what specific modal training data affects the final model performance results?

**Ethical Concerns:**

["NO or VERY MINOR ethics concerns only"]

**Final Justification:**

I sincerely appreciate the authors’ patient response. My concerns have been resolved, and I would recommend incorporating the above experiments and analyses into the final version. I have increased my score.

**Limitations:**

The computational resources consumed by the model are too large, the experiment lacks interpretability, only the number of frames used is compared in the data comparison, and not the impact of more factors such as training data on the experiment, more adequate experiments and the same amount of data are needed for comparison to make the paper more convincing.

**Paper Formatting Concerns:**

No major formatting issue

**Quality:**

3

**Strengths And Weaknesses:**

Strengths

The authors propose a groundbreaking adaptive weighting framework (Query-Based Connector) that dynamically reweights modalities under arbitrary missing inputs (vision/audio/speech), validated by TriSense-2M—the only dataset explicitly supporting modality dropout (2M samples), with best performance degradation under partial modality absence.

Weaknesses
1. The modal has computational efficiency concerns, the training demands are high: Stage 3 requires 16×A100 GPUs for 7 days, and also inference uses 64–128 frames per video, costing 8–10 hours for 11.4K test samples. While Slot-Based Compression reduces token counts, the LLM backbone and temporal modeling remain resource-intensive. No lightweight variants or quantization strategies are explored, hindering edge-device deployment.
2. Potential bias exists in the dataset. TriSense-2M relies on GPT-generated captions for its Generator or Judger training. Manual filtering (500 samples/batch) is sparse, potentially retaining inconsistent samples and the paper lacks analysis of bias impacts.
3. The modal in this paper underperforms in visual-only settings due to its multimodal optimization focus. This suggests the model prioritizes cross-modal synergy over single-modality refinement, limiting applications where audio/speech is unavailable. The authors acknowledge this but offer no mitigation strategy.

---

> ### Author Rebuttal · Authors · 2025-07-30
>
> **We thank all four reviewers for their constructive feedback:** "The paper is clearly written and easy to follow, with well-organized sections and clear explanations of both the method and experiments", "The introduction of the new TriSense 2M dataset is a valuable contribution to the community" (Reviewer ThZF); "This paper studies a very interesting problem which seeks to build an omni-modal large language model for comprehensive multi-modal understanding" (Reviewer 1DNL); "The authors propose a groundbreaking adaptive weighting framework validated by TriSense-2M" (Reviewer NaTe); "Make substantial data contributions, which are significant for omni-model exploration" (Reviewer 9tN8). **Next, we will carefully address each of the reviewers’ comments.**
>
> > Q1 & Q4: Training requires 16×A100 GPUs for 7 days; inference on 11.4K videos takes 8–10 hours (64–128 frames). Despite Slot-Based Compression, LLM and temporal modeling remain costly. No lightweight or quantized variants are offered. Table 4 lacks broader efficiency metrics.
>
> Stage 3 involves full fine-tuning a 7B VLM on 2M samples, so using 16×A100 GPUs for 7 days **is standard** to training times reported in recent video-LLM works [1,2]. After CUDA/driver upgrades, training now reduces to **~5 days.** We plan to explore lightweight variants (e.g., quantization) moving forward. Table 4 is to demonstrate that TriSense can achieve strong performance on public benchmarks with **fewer frames**, so it is only related to frames.
>
> To further clarify the computational efficiency, we conducted a comparison of inference performance against recent VLMs on a single H100 GPU using 64-frame input over 500 samples. Despite two additional modalities, our model achieves **competitive inference speed** and **outperforms** Qwen2.5-Omni with Flash Attention. Importantly, the inclusion of these modalities leads to significant omni-modal performance gains, thus a marginal trade-off in inference speed **is worthwhile**.
>
> | Model                         | Avg. CPU Time | Avg. GPU Time | Avg. Infer Time |
> | ----------------------------- | ------------- | ------------- | --------------- |
> | TimeChat                      | -             | -             | 1.33s           |
> | TRACE                         | 0.42s         | 1.02s         | 1.45s           |
> | Qwen2.5-Omni (w/o flash-attn) | 1.55s         | 6.21s         | 7.77s           |
> | Qwen2.5-Omni (w flash-attn)   | 1.58s         | 1.19s         | 2.78s           |
> | TriSense (Ours)               | 1.08s         | 1.16s         | 2.25s           |
>
> > Q2: TriSense-2M uses GPT-generated captions for training, risking bias. Sparse manual filtering (500 samples/batch) may let inconsistencies remain, with no analysis of bias impact provided.
>
> Since each event has four different caption versions and we set a **high acceptance threshold of 80%**, our actual workload was **4–5 times** higher than the basic sample count, so 500 samples per batch already resulted in a heavy workload for us. To overcome the potential bias, the pass rate for the judger was **set relatively low**. As mentioned at the end of **Section 3**, we started with **5 million** samples and ultimately selected only **2 million** after filtering.
>
> To further validate annotation quality, we conducted a **human evaluation** on 100 randomly selected samples generated by the generator. Each was **scored** by a human annotator who **has not seen the model output** according to the rules in Figure 5. Lastly, **91%** samples are passed by human. The results show that while human captions were often more fluent, model-generated captions showed **strong** semantic adequacy and scalability advantages. Due to text constraints, we cannot provide more samples here, but we have provided **scoring samples** in the **response of Q3 for Reviewer 1DNL.**
>
> > Q3: The model performs poorly in visual-only settings because it is optimized for multimodal synergy rather than single-modality refinement. This limits its applicability when audio or speech is missing. Although this limitation is acknowledged, no mitigation strategy is provided.
>
> The model achieves SOTA performance on **62.5%** of visual-only benchmarks, as reported in Section 5, indicating **strong competitiveness**. Therefore, the visual-only performance is **not degraded**, in the meantime, it is **also optimized for omni-modal flexibility**. To further strengthen visual-only capability, possible mitigation includes using a balanced ratio of omni-modal and visual-only samples (e.g., 90% and 10%), for which we are already taking actions. This may avoid overfitting to multimodal synergy while preserving single-modal ability.
>
> > Q5: Using closed-source GPT-o1 for Judger training limits reproducibility. It’s unclear if open-source LLMs were tested or how different LLMs affect the data.
>
> We would like to clarify that, similar to previous works [1,2], closed-source models such as GPT have been **widely adopted**, as their instruction-following and captioning capabilities are generally recognized as superior to those of open-source models. To further improve reproducibility, we have included all our prompts in Figure 5 of **Appendix B**. Given the strong instruction-following ability of GPT-o1, and more importantly, the incorporation of human review, we believe that reproducibility **should not be a concern**.
>
> To further evalute the generation quality of different models, we generated 100 omni-modal captions (AVS, VS, AV) with four different models and had human evaluators review each model’s outputs according to the standards in Figure 5. The results show that although open-source models like DeepSeek and Qwen3 can finish the task, their pass rates are **not as high as** closed models, and their outputs are **not sufficiently fluent or natural** in terms of English expressions. Therefore, it is necessary for us to **manually select high-quality data from closed-source models** and fine-tune the open-source models accordingly.
>
> | GPT-o1 (Close) | GPT-o4-Mini (Close) | DeepSeekR1-671B (Open) | Qwen3-235B (Open) |
> | -------------- | ------------------- | ---------------------- | ----------------- |
> | 92%            | 93%                 | 89%                    | 88%               |
>
> > Q6: In Section 3, have you calculated the percentage of missing modal data, and what specific modal training data affects the final model performance results?
>
> Approximately 10% of videos in TriSense-2M lack audio, in these cases, both audio and speech inputs are replaced with zero tensors. For the videos with audios, the model is forced to learn how to handle such dropout via dynamic modality reweighting, allowing robust performance under partial input conditions, as proven by our results in Section 5.
>
> To assess individual modality contributions, we conducted an ablation study using 20K samples over 3 epochs training the model **from scratch**, selectively removing branches of the Query-Based Connector, benchmarking on randomly selected 500 samples. For example, **AV-Only** in the table below means we discarded the Speech branch. We initialize the LLM from [1] to avoid potential biases in TriSense’s LLM towards omni-modal connector.
>
> Results show that all three modalities **contribute positively** to all tasks, with the full AVS configuration performing best. Visual-only and dual-modal variants perform reasonably well, confirming graceful degradation. Due to resource and time constraints, we are unable to conduct larger-scale experiments at this stage, but we plan to include more in future versions of the paper.
>
> | Model    | AVS-SC                                       | VS-SC                                        | AV-SC                                        | V-SC                                         |
> | -------- | -------------------------------------------- | -------------------------------------------- | -------------------------------------------- | -------------------------------------------- |
> | V-Only   | **B:** 1.0 **M:** 1.7 **R:** 10.1 **C:** 0.9 | **B:** 1.1 **M:** 1.6 **R:** 10.3 **C:** 1.1 | **B:** 1.0 **M:** 1.7 **R:** 10.2 **C:** 1.1 | **B: 1.0 M: 1.8 R:** 13.3 **C:** 1.4         |
> | VS-Only  | **B:** 1.7 **M:** 2.9 **R:** 13.3 **C: 2.4** | **B:** 1.6 **M:** 2.9 **R:** 14.7 **C: 2.5** | **B:** 0.7 **M:** 2.3 **R:** 13.2 **C:** 1.5 | **B:** 0.7 **M:** 1.6 **R:** 13.2 **C:** 1.2 |
> | AV-Only  | **B:** 1.7 **M:** 3.1 **R:** 13.7 **C:** 1.9 | **B:** 1.4 **M:** 2.2 **R:** 12.0 **C:** 2.1 | **B:** 0.9 **M:** 3.1 **R:** 15.0 **C: 1.7** | **B:** 0.7 **M:** 1.6 **R:** 13.0 **C:** 1.1 |
> | AVS-Full | **B: 1.8 M: 3.4 R: 14.6 C: 2.4**             | **B: 1.7 M: 3.1 R: 15.1 C: 2.5**             | **B: 1.0 M: 3.2 R: 15.3 C: 1.7**             | **B:** 0.8 **M: 1.8 R: 13.4 C: 1.5**         |
>
> | Model    | AVS-MR                        | VS-MR                         | AV-MR                         | V-MR                          |
> | -------- | ----------------------------- | ----------------------------- | ----------------------------- | ----------------------------- |
> | V-Only   | **R@0.5:** 0.1 **R@0.7:** 0.0 | **R@0.5:** 0.1 **R@0.7:** 0.0 | **R@0.5:** 0.1 **R@0.7:** 0.1 | **R@0.5: 0.4 R@0.7: 0.2**     |
> | VS-Only  | **R@0.5:** 0.4 **R@0.7:** 0.1 | **R@0.5: 0.3 R@0.7: 0.2**     | **R@0.5:** 0.1 **R@0.7:** 0.0 | **R@0.5:** 0.3 **R@0.7:** 0.1 |
> | AV-Only  | **R@0.5:** 0.4 **R@0.7:** 0.2 | **R@0.5:** 0.1 **R@0.7:** 0.1 | **R@0.5: 0.3** **R@0.7:** 0.1 | **R@0.5:** 0.3 **R@0.7:** 0.1 |
> | AVS-Full | **R@0.5: 0.4 R@0.7: 0.3**     | **R@0.5: 0.3 R@0.7: 0.2**     | **R@0.5: 0.3 R@0.7: 0.2**     | **R@0.5: 0.4 R@0.7: 0.2**     |
>
> **References:**
>
> [1] Guo, Yongxin, et al. "TRACE: Temporal Grounding Video LLM via Causal Event Modeling". ICLR, 2025.
>
> [2] Guo, Yongxin, et al. "VTG-LLM: Integrating timestamp knowledge into video llms for enhanced video temporal grounding". AAAI, 2025.

---

> > ### Comment · Reviewer_NaTe · 2025-08-06
> >
> > Thank you for the authors' efforts in addressing the rebuttal, which has resolved most of my concerns. Here are my further questions:
> >
> > - The annotated videos in VAST and InternVid are very short, while the videos in TriSense-2M are over 900 seconds long. Does this mean that for a 900-second video, there is only one moment of around 10 seconds with a caption, leading to extremely sparse temporal annotations? Therefore, it can only perform SC and MR tasks, but not dense video captioning.
> >
> > - The model compresses a very long video into just 16 tokens per modality, which results in a significant loss of information, including information about the target moment. Can the model really learn the localization capability? Has there been any attempt to retain more tokens?
> >
> > - Is it necessary to construct task-specific heads in the model? This could reduce the generalization ability of the large language model. Have there been attempts to use a unified head for multiple tasks?

---

> > > ### Author Response · Authors · 2025-08-06
> > > **Futher Clarification for Reviewer NaTe**
> > >
> > > Thank you for your follow-up question. We would like to provide the following clarifications:
> > >
> > > > The annotated videos in VAST and InternVid are very short, while the videos in TriSense-2M are over 900 seconds long. Does this mean that for a 900-second video, there is only one moment of around 10 seconds with a caption, leading to extremely sparse temporal annotations? Therefore, it can only perform SC and MR tasks, but not dense video captioning.
> > >
> > > **Sparse temporal annotations:** The annotations released by VAST and InternVid correspond to different events within the same video, for example, video `abc` may include captions for `event1`, `event2`, `event3`, etc. Therefore, many of the annotated videos in these two datasets are **not short**, because they have large number of events, and the longest videos can even **exceed one hour** in duration. When selecting videos for TriSense-2M, we intentionally leveraged this feature to prioritize longer videos, as they present greater challenges for model understanding. We also analyzed the number of events per video in TriSense-2M, as shown in the following table, which demonstrates that **TriSense-2M’s temporal annotations are not sparse**:
> > >
> > > |                      | Min  | Max  | Average |
> > > | -------------------- | ---- | ---- | ------- |
> > > | **Number of Events** | 8    | 167  | 51      |
> > >
> > > **Dense video captioning (DVC):** We previously attempted to perform DVC tasks using TriSense-2M. However, the large number of events per video resulted in excessively long annotations, significantly increasing the **text token length** during training. Even after compressing the number of multimodal tokens, our computational resources were insufficient to complete this task.
> > >
> > > However, based on the discussion above, because TriSense-2M has the features of **long video durations** and **dense event coverage**, we believe the dataset holds strong potential for supporting DVC tasks. This is also an important direction of our future work.
> > >
> > > > The model compresses a very long video into just 16 tokens per modality, which results in a significant loss of information, including information about the target moment. Can the model really learn the localization capability? Has there been any attempt to retain more tokens?
> > >
> > > **16 tokens per modality:** We would like to clarify that our approach does **not** compress the entire video into just 16 tokens per modality. Instead, **each frame and its corresponding audio segment** are compressed into 16 tokens. For example, when sampling 64 frames, the feature shape of each modality becomes $64$ frames × $16$ tokens of 4096 dimensions, i.e., **[$64$, $16$, $4096$].** This results in **1024 tokens per modality** ($64 \times 16$). Increasing the number of sampled frames, such as to 128 frames, would raise the number of tokens to **2048 per modality.**
> > >
> > > Lastly, we would like to emphasize that modality compression is a **common** solution. Prior work [1] has shown that even with more aggressive compression (e.g., down to 8 tokens), models can still achieve strong performance.
> > >
> > > **Attempt to retain more tokens:** In fact, slot compression is a highly flexible component. We can easily reduce the compression ratio, for example, by setting the number of tokens per frame to 32, 64, or even 128. Since the primary factor affecting memory usage is the **total number of tokens** (multimodal + textual), we currently set the per-frame token count to 16. This allows us to use a **large batch size** (e.g., 128 or 256) during training, which helps reduce the risk of overfitting.
> > >
> > > Looking ahead, we plan to train a lower-compression version, such as 128 frames with 32 tokens per frame using a smaller batch size to explore more potential.
> > >
> > > > Is it necessary to construct task-specific heads in the model? This could reduce the generalization ability of the large language model. Have there been attempts to use a unified head for multiple tasks?
> > >
> > > **Task-specific heads and generalization ability:** According to prior work [1] and our VideoMME results in Appendix Table 10, task-specific heads not only enable strong performance in temporal understanding but also help preserve general understanding capabilities. Our primary motivation for introducing the time head is to enhance the model’s temporal reasoning by **incorporating time information** alongside the LM head, but note that the LM head is still the primary modeling component.
> > >
> > > **Attempts to use a unified head for multiple tasks:** We agree that more unified modeling approaches may emerge in the future, for example, injecting temporal information via positional encodings or explicitly prompting the model. We are already actively exploring these possibilities.
> > >
> > > **References:**
> > >
> > > [1] Guo, Yongxin, et al. "TRACE: Temporal Grounding Video LLM via Causal Event Modeling". ICLR, 2025.

---

> > > > ### Comment · Reviewer_NaTe · 2025-08-07
> > > >
> > > > Thank you for the author's response, which has resolved my concerns. Below are some suggestions:
> > > >
> > > > - I recommend expanding Table 1 and providing more detailed data distribution charts, including the total number of events, average event duration, event duration distribution, and the distribution of the ratio of event duration to video length, for a deeper understanding of the dataset.
> > > >
> > > > - I noticed that no information has been provided about the test set used, such as the number of videos, the number of events, and the distribution. I suggest adding a detailed description of the test set.
> > > >
> > > > - I acknowledge that completing model training within a limited time is challenging, but I still hope that the full version of the paper will include ablation studies on the heads and token compression rates to further enhance the paper.
> > > >
> > > > - I sincerely hope the authors can release the training and testing data to contribute to the open-source community.

---

> > > > > ### Author Response · Authors · 2025-08-08
> > > > > **Further Responses to Reviewer NaTe (1/2)**
> > > > >
> > > > > Thank you for your valuable suggestions, which have greatly contributed to improving the quality of our paper! Although time was limited, we have made meaningful revisions, including clarifications, additions, and quantitative updates wherever possible. We respectfully hope that the revisions we have made will be taken into account in your overall assessment:
> > > > >
> > > > > > I recommend expanding Table 1 and providing more detailed data distribution charts, including the total number of events, average event duration, event duration distribution, and the distribution of the ratio of event duration to video length, for a deeper understanding of the dataset.
> > > > >
> > > > > In the final version of the paper, we will provide additional details to enhance the reader’s understanding of our work. This will include, but is not limited to, Table 1: Number of Events / Duration, and Table 2: Distribution of Event Durations.
> > > > >
> > > > > | Total Events | Avg. Duration | Min Duration | Max Duration |
> > > > > | ------------ | ------------- | ------------ | ------------ |
> > > > > | 1940522      | 6.87s         | 2.00s        | 30.00s       |
> > > > >
> > > > > | 0-5s           | 5-10s          | 10-15s         | 15-20s        | 20-30s        |
> > > > > | -------------- | -------------- | -------------- | ------------- | ------------- |
> > > > > | 689842 (35.5%) | 896501 (46.2%) | 244065 (12.6%) | 81173 (4.2 %) | 289626 (1.5%) |
> > > > >
> > > > > In Table 1, **Avg. Duration** refers to the average length of all events, while **Min/Max Duration** represents the durations of the shortest and longest events, respectively. Table 2 presents the number and proportion of events falling within predefined duration ranges (e.g., 0–5s, 5–10s, etc.).
> > > > >
> > > > > The ratio of event duration to video length is best illustrated through a visual chart. However, since images cannot be included at this rebuttal stage, we will provide this visualization in the **final version** of the paper. Additionally, we plan to incorporate further descriptive statistics, such as the standard deviation and median of event durations, to give a more comprehensive overview.
> > > > >
> > > > >
> > > > >
> > > > > > I noticed that no information has been provided about the test set used, such as the number of videos, the number of events, and the distribution. I suggest adding a detailed description of the test set.
> > > > >
> > > > > As noted in the last paragraph of **Appendix A.2**, we selected 11,415 challenging samples to construct our test set. Each sample includes eight distinct tasks, including four different combinations each for Segment Captioning and Moment Retrieval. We will provide additional detailed information to enhance transparency and reproducibility in the revised supplementary materials, **including but not limited to:**
> > > > >
> > > > > | Num. Videos | Num. Events | Avg. Event Duration | Total Tasks |
> > > > > | ----------- | ----------- | ------------------- | ----------- |
> > > > > | 3805        | 11415       | 7.23s               | 91320       |

---

> > > > > > ### Author Response · Authors · 2025-08-08
> > > > > > **Further Responses to Reviewer NaTe (2/2)**
> > > > > >
> > > > > > > I acknowledge that completing model training within a limited time is challenging, but I still hope that the full version of the paper will include ablation studies on the heads and token compression rates to further enhance the paper.
> > > > > >
> > > > > > Although time was limited, we conducted a series of small-scale experiments to help clarify our approach. Specifically, we randomly selected 20K samples and trained for 3 epochs, initializing all weights from [1]. All experiments were conducted with a sampled frame rate of 64.
> > > > > >
> > > > > > For the prediction head, we **removed** the Time Head and instead encoded temporal information into special tokens, which were injected into the LLM. Prediction was then performed using only the LM Head. In terms of token compression, we explored different slot configurations, including **8, 32, and 64.**
> > > > > >
> > > > > > |         | AVS-SC                                   | AVS-MR                    |
> > > > > > | ------- | ---------------------------------------- | ------------------------- |
> > > > > > | Slot=8  | B: 1.6 M: 3.1 R: 13.1 C: 1.9             | R@0.5: 0.3 R@0.7: 0.1     |
> > > > > > | Slot=16 | B: 1.8 M: 3.4 R: 14.6 **C: 2.4**         | R@0.5: 0.4 **R@0.7: 0.3** |
> > > > > > | Slot=32 | B: 1.8 M: 3.4 R: 15.6 **C: 2.4**         | R@0.5: 0.4 **R@0.7: 0.3** |
> > > > > > | Slot=64 | **B: 1.9** **M: 3.5 R: 16.1** **C: 2.4** | **R@0.5: 0.5** R@0.7: 0.2 |
> > > > > >
> > > > > > |                     | AVS-SC                           | AVS-MR                    |
> > > > > > | ------------------- | -------------------------------- | ------------------------- |
> > > > > > | Unified Head        | B: 1.5 M: 3.1 R: 13.2 C: 1.1     | R@0.5: 0.2 R@0.7: 0.1     |
> > > > > > | Time Head + LM Head | **B: 1.8 M: 3.4 R: 14.6 C: 2.4** | **R@0.5: 0.4 R@0.7: 0.3** |
> > > > > >
> > > > > > Experiments show that:
> > > > > >
> > > > > > 1. **Slot Compression:** Although reducing the compression rate (by increasing the number of slots) can bring **slight performance improvements,** it also results in a **significant increase in computational overhead.** Therefore, the benefits are subject to diminishing returns. As shown in the table, using slot=16 currently offers a good balance between performance and computational cost.
> > > > > > 2. **Head:** Using a single unified head alone leads to a **significant drop** in performance, indicating that the time head provides additional temporal information which is beneficial to the model. There may be better unified solutions in the future.
> > > > > >
> > > > > >
> > > > > >
> > > > > > > I sincerely hope the authors can release the training and testing data to contribute to the open-source community.
> > > > > >
> > > > > > As emphasized in the abstract, all components of TriSense-2M, including the dataset and training/testing scripts, will be released to the open-source community. To ensure proper use for academic research, we will adopt a **non-commercial** open-source license along with an **Acceptable Use Policy (AUP).**
> > > > > >
> > > > > > We believe TriSense-2M will serve as **a valuable pretraining resource** for the academic community. Additionally, the TriSense model will be released with robust open-source weights, enabling researchers to **build upon our pretrained models** and more efficiently pursue promising research directions.

---

> ### Comment · Area_Chair_698r · 2025-08-05
> **post-rebuttal comments**
>
> Dear reviewer NaTe,
>
> As you may have seen already, the authors have responded to the questions in your initial review. Can you please share your thoughts post rebuttal, once you've had a chance to see the author responses and also the other reviews?
>
> Best, AC

---

### Official Review · Reviewer_1DNL · 2025-07-03

**Clarity:** 4
**Significance:** 4
**Originality:** 3
**Rating:** 4
**Confidence:** 4

**Summary:**

This paper introduces a novel triple-modality large language model for detailed video understanding that considers visual, audio and speech modalities. The proposed model incorporates a query-based connector which can automatically fuse multi-modal features. Besides, the authors collect a large-scale dataset comprising 2 million samples with synchronized video, audio and speech. The experimental results and the downstream applications demonstrate the effectiveness of the method and dataset.

**Questions:**

1. Could you provide a detailed explanation of the feature shapes for the vision, audio, and speech modalities? Additionally, how many tokens are ultimately fed into the large language model?

**Ethical Concerns:**

["NO or VERY MINOR ethics concerns only"]

**Final Justification:**

The rebuttal well addressed my concerns and I remain positive about the paper.

**Limitations:**

No. In line 273, the authors find that “model shows slightly lower performance on visual-only moment retrieval compared to state-of-the-art vision models”, future work could explore methods for balancing the multi-modality question answering performances.

**Quality:**

3

**Strengths And Weaknesses:**

**Strength:**
1. This paper studies a very interesting problem which seeks to build an omni-modal large language model for comprehensive multi-modal understanding.
2. The proposed TriSense-2M dataset is useful and contains various types of captions including AVS, AV, VA captions, which can support a wide range of future studies.
3. The proposed query-based connector in the TriSense model is novel, as it extends the previous Q-former-based connector to support three modalities: audio, visual, and speech for multi-modal tasks.

**Weakness:**
1. The overall quality of the proposed TriSense-2M dataset is not very clear. Could you provide some randomly picked examples from the dataset?
2. According to the recommendation of the NeurIPS conferences, the authors should discuss the limitation of their proposed models and potential future improvements.
3. As mentioned in Appendix B, in the dataset construction pipeline, only 10,000 and 3,000 samples are used to train the generator and the judger, respectively. This raises my concerns about their captioning and judging capabilities for reliable data annotation.

---

> ### Author Rebuttal · Authors · 2025-07-27
>
> **First of all, we would like to sincerely thank the four reviewers for their constructive feedback on our manuscript:** "The paper is clearly written and easy to follow, with well-organized sections and clear explanations of both the method and experiments", "The introduction of the new TriSense 2M dataset is a valuable contribution to the community" (Reviewer ThZF); "This paper studies a very interesting problem which seeks to build an omni-modal large language model for comprehensive multi-modal understanding" (Reviewer 1DNL); "The authors propose a groundbreaking adaptive weighting framework validated by TriSense-2M" (Reviewer NaTe); "Make substantial data contributions, which are significant for omni-model exploration" (Reviewer 9tN8). **Next, we will carefully address each of the reviewers’ comments.**
>
> > Q1: The overall quality of the proposed TriSense-2M dataset is not very clear. Could you provide some randomly picked examples from the dataset.
>
> Due to NeurIPS and OpenReview rebuttal policies, we are **unable** to include any visual samples or external links at this stage. However, we fully acknowledge the importance of illustrating dataset quality through representative examples, we have also included a certain number of image samples in **Appendix D** of the initial version of the manuscript. In the meanwhile, to address this, we provide qualitative examples and evaluations in our **response to Q3**, where we include a selection of text-based samples spanning multiple modality combinations (audio, visual, and speech). These include source captions, generated captions, human-written references, and judger scores. To further promote transparency and reproducibility, we will release the code and data on GitHub.
>
> > Q2: According to the recommendation of the NeurIPS conferences, the authors should discuss the limitation of their proposed models and potential future improvements.
>
> Below, we outline the current limitations of our work and future directions for improvement from two perspectives:
>
> 1. **Dataset Quality:** While extensive filtering reduced the dataset from 5M to 2M samples, full manual verification at this scale remains challenging. Moving forward, we anticipate further quality improvements through integration of stronger models such as GPT-o3 or Gemini 2.5 Pro. Additionally, selectively expanding the volume of human-reviewed samples and scaling the dataset to 5–10M entries without compromising quality are promising avenues that will enhance the dataset’s utility in both pretraining and transfer learning tasks.
> 2. **Model Design (Query-Based Connector):**  Potential refinements include the adoption of 2D-RoPE in place of absolute positional encodings, the substitution of LayerNorm with RMSNorm for improved stability, and the incorporation of activation functions like SwiGLU, which have shown efficacy in large language models. Moreover, our ongoing work includes optimizing the dynamic modality weighting mechanism to further strengthen multimodal alignment and representation fidelity. We are already implementing these improvements and will open-source them on GitHub.
>
> Lastly, while we acknowledge the limitations, we believe that **the contributions of this paper outweigh its limitations.** These limitations are not fundamental flaws but rather natural outcomes of working at the frontier of model and dataset development. We view them as opportunities that will enrich the future evolution. The proposed **TriSense-2M** dataset is expected to serve as a **strong pretraining foundation** for large multimodal models. Furthermore, the **TriSense** model trained on this dataset can act as a **powerful foundation model** for future research, supporting fine-tuning and further improvements while significantly reducing training costs. To promote reproducibility and encourage continued development, we will release the full codebase, including the training pipeline and connector implementation on GitHub.
>
> > Q3: As mentioned in Appendix B, in the dataset construction pipeline, only 10,000 and 3,000 samples are used to train the generator and the judger, respectively. This raises my concerns about their captioning and judging capabilities for reliable data annotation.
>
> The 10,000 and 3,000 samples refer to manually reviewed sets, however, due to the multiple-caption-per-sample design (AVS, AV, VS), the actual review workload was **4–5× larger**, which is already a heavy workload for us. To overcome the potential bias, we applied a strict acceptance threshold (**80%**) and set a **low pass rate** for the judger model, resulting in **rigorous filtering.** Out of 5M initial samples, only 2M were retained.
>
> To further validate annotation quality, we conducted a **human evaluation** on 100 randomly selected samples generated by the generator. Each was **scored** by a human annotator who **has not seen the model output** according to the criterias in Figure 5. Lastly, **91/100** samples are passed by human, yielding a **91%** acceptance rate. The results show that while human captions were often more fluent, model-generated captions showed **strong** semantic adequacy and scalability advantages. Due to text constraints, we cannot show all of them to you, but we do our best to show you some in the table below, and we will include more samples in the future version.
>
> | Source Caption                                               | Generated Caption                                            | Human Caption                                                | Judger Score | Human Rate |
> | ------------------------------------------------------------ | :----------------------------------------------------------- | :----------------------------------------------------------- | ------------ | ---------- |
> | **audio:** a person is opening and closing an umbrella. **visual:** outside a woman is showing how to decorate a christmas tree. **speech:** So I thought these would sit up on the desk too and just be really pretty and neutral. So all of that came from Kirklands. | **avs:** A woman shows how to decorate a Christmas tree outside while a voice of opening and closing an umbrella is heard, and the woman thinks the decorations will look nice on a desk and be neutral in color, all from Kirklands. **av:** A woman is outside decorating a Christmas tree, and a person is opening and closing an umbrella somewhere else. **vs:** A woman sits outside decorating a Christmas tree and showing art she has finished, mentioning how pretty and nice they would look. | **avs:** A woman is decorating a Christmas tree outdoors. While decorating, she introduces to the audience that these decorations are from Kirklands, which can bring a pretty and neutral visual effect. Meanwhile, someone in the background is opening and closing an umbrella. **av:** A woman is decorating a Christmas tree outdoors. At the same time, someone in the background is opening and closing an umbrella. **vs:** A woman is decorating a Christmas tree outdoors. While decorating, she introduces to the audience that the decorations are from Kirklands, which can bring a pretty and neutral visual effect. | 5            | 4          |
> | **audio:** someone is messing with a cup. **visual:** a woman is showing a drawer full of lipstick. **speech:** So those are my lip balms and then these are all lip glosses. I have a lip tar, so thats, not the little car. | **avs:** A female is showing of her drawer full of various kinds of lipstick, including lip balms and lip glosses, and she mentions that she has a lip tar, messing with a cup in the meantime. **av:** A woman is messing with a cup and showing her drawer, which is full of different kinds of lipstick. **vs:** A girl is talking about her makeup drawer, mentioning various lip balms, lip glosses, and other beauty products, including a lip tar. | **avs:** A girl is showing a drawer filled with different lipsticks in the video. While introducing the various types of her beauty products, she is also messing with a cup. **av:** A girl is showing a drawer filled with different lipsticks in the video, while messing with a cup. **vs:** A girl is showing a drawer filled with different lipsticks in the video, and at the same time, she is introducing the various types of her beauty products. | 4            | 4          |
>
> > Q4: Could you provide a detailed explanation of the feature shapes for the vision, audio, and speech modalities? Additionally, how many tokens are ultimately fed into the large language model?
>
> As described in Section 4.1, we use slot compression to reduce the number of tokens for each modality to 16. If we set the number of frames to 64, then for the **vision, audio, and speech** modalities, the feature shape will be $64$ frames $\times$ $16$ tokens of 4096 dimensions, i.e., [$64$, $16$, $4096$]. The **query** typically consists of 40–50 tokens, i.e., [$40$, $4096$]. After inputting these four types of features into the Query Based Connector, it outputs a multimodal feature of [$1024$, $4096$]. If the number of sampled frames is set to 128, then the modality features will be [$128$, $16$, $4096$], and the final multimodal feature will increase to [$2048$, $4096$].

---

> > ### Comment · Reviewer_1DNL · 2025-08-06
> >
> > Thanks for the authors' response. All of my concerns have been addressed and I maintain my positive rating.

---

> > > ### Author Response · Authors · 2025-08-06
> > >
> > > Dear reviewer 1DNL,
> > >
> > > We sincerely appreciate your efforts, your valuable suggestions have helped us improve the quality of our paper.

---

> ### Comment · Area_Chair_698r · 2025-08-05
> **post-rebuttal discussion**
>
> Dear reviewer 1DNL,
>
> As you may have seen already, the authors have responded to the questions in your initial review. Can you please share your thoughts post rebuttal, once you've had a chance to see the author responses and also the other reviews?
>
> Best, AC

---

### Official Review · Reviewer_ThZF · 2025-07-03

**Clarity:** 2
**Significance:** 3
**Originality:** 2
**Rating:** 4
**Confidence:** 3

**Summary:**

This paper introduces TriSense, a multimodal large language model architecture designed for comprehensive video temporal understanding, integrating visual, audio, and speech modalities. The core novelty is a Query-Based Connector that adaptively weights and fuses modalities based on the input query, enhancing robustness to missing or irrelevant modalities. The authors also present TriSense-2M, a curated, large-scale dataset of over 2 million annotated long-form video segments incorporating variable modality combinations. Extensive evaluations across segment captioning and moment retrieval tasks on both TriSense-2M and several public benchmarks show that TriSense outperforms prior models, notably excelling in settings with partial modality availability. Ablation studies further substantiate the contributions of the adaptive connector and dataset design.

**Questions:**

My main concern is the novelty of the modeling, as I do not see substantial innovation in the design. But I appreciate the introduction of a high-quality AVS dataset, which is a valuable contribution to the community. Overall, I prefer to be borderline; but, since a strictly neutral rating is not available, I lean slightly positive and therefore give a borderline accept. I will make my final decision after reading the authors’ rebuttal.

**Ethical Concerns:**

["NO or VERY MINOR ethics concerns only"]

**Final Justification:**

The author's rebuttal on novelty is clear. Initially, I felt the work was borderline but leaned toward a borderline accept due to the lack of a neutral rating option. With the clarification provided, I now believe the current score is appropriate and will maintain my positive rating.

**Limitations:**

yes

**Paper Formatting Concerns:**

good

**Quality:**

3

**Strengths And Weaknesses:**

Strengths
* The paper is clearly written and easy to follow, with well-organized sections and clear explanations of both the method and experiments.
* The idea of a Query Based Connector to dynamically reweight contributions from different modalities is intuitive.
* The introduction of the new TriSense 2M dataset is a valuable contribution to the community.
* The proposed method achieves significant performance gains over strong baselines across multiple benchmarks, demonstrating its practical effectiveness.

 Weaknesses
* Limited novelty in modality fusion techniques. While the Query Based Connector is positioned as the key modeling innovation of the paper, its design closely resembles existing approachesp[1,2]. Specifically, the query based cross attention mechanism is highly similar to the instruction-aware QFormer proposed in InstructBLIP and X-InstructBLIP[1]. In addition, the dynamic reweighting of modalities resembles the self-gated query fusion strategy used in CREMA[2]. As such, the proposed connector appears to be more of a combination of these two prior techniques rather than a fundamentally novel modality fusion method. This overlap limits the originality of the technical contribution.
* Missing citations and related work discussion. Given that the Query Based Connector is the central technical innovation of the paper and even emphasized in the abstract, the lack of a thorough discussion of related work on modality fusion is a notable weakness. A more comprehensive related work section discussing these and clarifying how the proposed connector differs or improves would strengthen the paper and help position it more clearly in the landscape of multi modal fusion research.

References
\[1] X InstructBLIP: A Framework for Aligning Image 3D Audio Video to LLMs and its Emergent Cross modal Reasoning 2023
\[2] CREMA: Generalizable and Efficient Video Language Reasoning via Multimodal Modular Fusion ICLR 2025

---

> ### Author Rebuttal · Authors · 2025-07-30
>
> **First of all, we would like to sincerely thank the four reviewers for their constructive feedback on our manuscript:** "The paper is clearly written and easy to follow, with well-organized sections and clear explanations of both the method and experiments", "The introduction of the new TriSense 2M dataset is a valuable contribution to the community" (Reviewer ThZF); "This paper studies a very interesting problem which seeks to build an omni-modal large language model for comprehensive multi-modal understanding" (Reviewer 1DNL); "The authors propose a groundbreaking adaptive weighting framework validated by TriSense-2M" (Reviewer NaTe); "Make substantial data contributions, which are significant for omni-model exploration" (Reviewer 9tN8). **Next, we will carefully address each of the reviewers’ comments.**
>
> > Q1: Limited novelty in modality fusion techniques. While the Query Based Connector is positioned as the key modeling innovation of the paper, its design closely resembles existing approaches[1,2]. Specifically, the query based cross attention mechanism is highly similar to the instruction-aware QFormer proposed in InstructBLIP and X-InstructBLIP[1]. In addition, the dynamic reweighting of modalities resembles the self-gated query fusion strategy used in CREMA[2].
>
> We would like to respectfully claim that the proposed Query-Based Connector **is NOT** the combination of these techniques, with detailed explanations below:
>
> - **Architectural differences:**
>   1. **QFormer [1,2,3,4]** uses **fixed, learnable query vectors** to perform cross-attention across modalities, generating instruction-aware, fixed-length representations. In contrast, our method uses the **actual user-input prompt** as the query, making it flexible and **only used in the initial stage.** Because the cross-attention facilitates cross-modal interaction, placing the query alongside the target modality is a **natural and intuitive design choice.** Therefore, despite similarities, our approach **differs significantly** from QFormer in both the nature of the query and our overall objectives, as cross-attention is not the central contribution of our work.
>   2. **Self-gated Multimodal Query Fusion [4]:** In CREMA, video is treated as the **primary modality**, while other modalities are linearly compressed as **supportive modalities**, gated using a sigmoid function, and fused into a fixed-length token alongside the primary modality. In contrast, our Query-Based Connector treats all modalities equally, without designating any as primary or supportive, and **does not aim to align them to the primary modality.** Therefore, our proposed Query-Based Connector is **totally different** from Self-gated Multimodal Query Fusion from both architecture and concept.
>   3. **Proposed Query-Based Connector:** The query attends to each modality (visual, audio, and speech) independently via cross-attention to extract query-relevant features, this is only an **initial step** that is similar to QFormer, but **only superficially.** Next, all modalities are processed through an MLP followed by a softmax layer to generate dynamic weights conditioned on the query. This enables adaptive modality reweighting and temporally aligned fusion, which naturally suppresses missing modalities. **Unlike QFormer, our method goes beyond basic query-to-modality cross-attention. Unlike CREMA, we do not designate a major modality or attempt to merge other modalities into the major one. Notably, none of the referenced works incorporate modality reweighting.**
> - **Design Intentions Difference:** The primary goal of QFormer is to extract **single-modality** features to mitigate the modality gap. In contrast, CREMA’s self-gated Multimodal Query Fusion [4] is designed to **prevent the query token size from scaling linearly** with the number of modalities. Our Query-Based Connector, however, is built with a different design philosophy: to offer a **universal, simple, and effective** mechanism for extracting features from all modalities while enabling **dynamic weighting and fusion.** This approach naturally favors a concise connector architecture to maintain generalizability, avoid overfitting, and reduce optimization challenges
>
> Lastly, we want to respectfully emphasize that the connector itself is **carefully designed,** incorporating design elements such as inter-frame and inter-token positional encoding, the placement of LayerNorm, and the adjustment of softmax temperature—all of which contribute to its effectiveness in omni-modal settings. Our key contribution lies in demonstrating that such a concise and adaptable architecture can perform well across diverse modalities, a capability that, to our knowledge, **has not been shown in prior VLMs.** We believe this offers meaningful insights for future research. The proposed TriSense-2M dataset is intended to serve as a **robust pretraining foundation** for large multimodal models. Additionally, the TriSense model trained on this dataset can act as a **strong foundation model** for future studies, supporting fine-tuning and enhancement while substantially reducing training costs.
>
> **References:**
>
> [1] Dai, Wenliang, et al. "Instructblip: Towards general-purpose vision-language models with instruction tuning." *Advances in neural information processing systems* 36 (2023): 49250-49267.
>
> [2] Panagopoulou, Artemis, et al. "X-instructblip: A framework for aligning image, 3d, audio, video to llms and its emergent cross-modal reasoning." *European Conference on Computer Vision*. Cham: Springer Nature Switzerland, 2024.
>
> [3] Zhang, Hang, Xin Li, and Lidong Bing. "Video-LLaMA: An Instruction-tuned Audio-Visual Language Model for Video Understanding." *Proceedings of the 2023 Conference on Empirical Methods in Natural Language Processing: System Demonstrations*. 2023.
>
> [4] Yu, Shoubin, Jaehong Yoon, and Mohit Bansal. "CREMA: Generalizable and Efficient Video-Language Reasoning via Multimodal Modular Fusion." *The Thirteenth International Conference on Learning Representations*. 2024.
>
> > Q2: Missing citations and related work discussion. Given that the Query Based Connector is the central technical innovation of the paper and even emphasized in the abstract, the lack of a thorough discussion of related work on modality fusion is a notable weakness. A more comprehensive related work section discussing these and clarifying how the proposed connector differs or improves would strengthen the paper and help position it more clearly in the landscape of multi modal fusion research.
>
> The discussion of the Query-Based Connector in relation to existing connectors was concise in the original submission. We have now provided a detailed comparative analysis addressing this point in your **last concern** and will incorporate it into **future revisions** to further clarify the distinctions and strengthen the paper’s contribution.

---

> > ### Comment · Reviewer_ThZF · 2025-08-06
> > **Thanks for the rebuttal**
> >
> > I thank the reviewer for the clarification regarding the novelty, which largely addresses my initial concern. Initially, I felt the work was borderline but leaned toward a borderline accept due to the lack of a neutral rating option. With the clarification provided, I now believe the current score is appropriate and will maintain my positive rating.

---

> ### Comment · Area_Chair_698r · 2025-08-05
> **post-rebuttal comments**
>
> Dear reviewer ThZF,
>
> As you may have seen already, the authors have responded to the questions in your initial review. Can you please share your thoughts post rebuttal, once you've had a chance to see the author responses and also the other reviews?
>
> Best, AC

---

### Note · Authors · 2025-08-12

We thank the reviewers and ACs for their thoughtful feedback and constructive guidance throughout the rebuttal process.


Initially, our paper received three positive scores and one negative score. The main concerns focused on the novelty of the Query-Based Connector, more details about the dataset, and ablation studies across different modalities. In response, we provided clarifications and comprehensive dataset information, such as event type distribution and event length, and we also supplied as many additional detailed experiments as possible to address the reviewers’ concerns. At the end, we received positive reviews from all four reviewers. We are pleased that the reviewers found our rebuttal to have addressed their concerns, and we are grateful for the increase in the final score.
In the future version, we will include all experiments discussed in the rebuttal and commit to releasing TriSense-2M and TriSense to the open-source community, contributing to the academic field. We believe this paper represents a valuable contribution to the field.


Once again, we are grateful for the time, care, and service of the reviewers, ACs, SACs, and PCs in shepherding this submission.

---

### Decision · Program_Chairs · 2025-09-17

**Decision:**

Accept (poster)

**Comment:**

This paper makes two broad contributions: (1) a large multimodal model TriSense for video understanding that leverages audio, visual and speech modalities, (2) a dataset TriSense-2M built from existing benchmarks based on YouTube videos. In terms of the core technical contribution, the paper proposes a query-based connector that weights and thus fuses modalities based on their importance and their availability. Such "inequalities" in video data are also relfected in the proposed dataset. TriSense is evaluated on TriSense-2M as well as several standard public benchmarks to show its effectiveness.

All the reviewers and the AC appreciated the strengths of the paper, which include an approach to leverage multiple modalities for video understanding tasks and also an interesting dataset that is relevant for the research community. However, several questions on the novelty, relation to other fusion-based approaches, the quality of the proposed dataset, computational efficiency, potential biases, were noted in the initial reviews. The rebuttal and the discussion phases clarified all these concerns, resulting in the reviewers's overall positive evaluation of this paper. Thus, the AC recommends the paper for acceptance. The clarifications provided in the rebuttal and during the discussion phase should be included in the final version of the paper. The final version should also carefully address the ethical questions.